

# DAR-type model based on "long memory-threshold" structure: a competitor for daily streamflow prediction under changing environment

Huimin Wang[1,2], Songbai Song[3,4*], Zhuoyue Peng[1], Gengxi Zhang[1*]

[1]College of Hydraulic Science and Engineering, Yangzhou University, Yangzhou 225009, China
[2]Modern Rural Water Resources Research Institute, College of Hydraulic Science and Engineering, Yangzhou University, Yangzhou 225009, China
[3]College of Water Resources and Architectural Engineering, Northwest A & F University, Yangling 712100, China
[4]Key Laboratory for Agricultural Soil and Water Engineering in Arid Area of Ministry of Education, Northwest A & F University, Yangling 712100, China

*Correspondence to*: Songbai Song (ssb6533@nwafu.edu.cn) and Gengxi Zhang (gengxizhang@yzu.edu.cn)

**Abstract.** The non-stationarity, non-linearity, and time-varying fluctuations of streamflow have increased with changes in the environment, challenging accurate streamflow prediction. Furthermore, the overlook of long-term memory features could lead to biases in model parameter estimation and testing of time series properties. The classical linear Autoregressive-Generalized Autoregressive Conditional Heteroskedasticity (AR-GARCH) model has a narrow parameter range, and the moment conditional requirements for parameter estimation are relatively strict, limiting its applicability and prediction accuracy in modelling and predicting daily streamflow. Under the premise of long-term memory, a dual-threshold double autoregressive (DTDAR) model is proposed to capture the non-linear patterns in streamflow series. Using 15 hydrological stations in the Yellow River basin in China as an example, DAR models are compared with AR-GARCH models to assess their applicability and predictive ability. The results indicate that the DAR-type models have a stronger predictive ability for daily streamflow than the AR-GARCH-type models. The threshold models (DTDAR, TAR-GARCH) convert non-linear transformations into several linear problems, improving the prediction accuracy of single linear structural models (DAR and FDAR, AR-GARCH and FAR-HARCH), among which the $R^2$ value is improved by 29.15% and 15.06%, 25.53% and 15.53%, and the NSE value is increased by 0.29 and 0.16, 0.24 and 0.15. Compared to the normal distribution, the student's t distribution for residuals is a better choice for predicting daily streamflow time series in the study area. This study enriches the stochastic hydrological models and improves the accuracy of streamflow prediction.

## 1 Introduction

The behavioural characteristics of hydrological systems have become increasingly complex in the face of climate change and human activities (Lyu et al., 2023; Ma et al., 2024; Matic et al., 2022; Sivakumar, 2009). The hydrological statistical method has gained wide attention in hydrological simulation and prediction due to its outstanding ability to describe the structure,



function, and development trend of the system, and has become one of the hot issues in stochastic hydrology research (Can et al., 2012; Chen et al., 2021; Yang et al., 2022). As such, several stochastic models have been proposed, among which the regression model has a simple form and is easy to implement.

Traditional regression models, including the autoregressive (AR) model, the moving average (MA) model, and their combined form (ARMA model), are well-suited for stationary time series but have significant limitations. For non-stationary series, the differencing ARMA (ARIMA) model is an effective means, and its key difference from the ARMA model is the addition of a difference step to smooth the sequence (Wang et al., 2019). However, both the ARIMA and ARMA models struggle to handle seasonality in time series. The seasonal ARIMA (SARIMA) model is the preferred option for monthly streamflow time series with annual cycles resulting from seasonal variations (Adnan et al., 2019; Modarres, 2007). However, the illegible seasonal features in the daily streamflow series bring a challenge to the applicability of the SARIMA model. To overcome this issue, Guo et al. (2021a) used seasonal standardization to preprocess daily streamflow time series, removing the influence of seasonality for subsequent research. So, this study uses the seasonal standardization method to eliminate the seasonal effect from daily streamflow time series.

In the last decades, the conditional heteroscedastic behaviour (ARCH effect) of the mean model residual series obtained increasing attention due to changing environment (Guo et al., 2021b; Nazeri-Tahroudi et al., 2022; Wang et al., 2023). The generalized Autoregressive Conditional Heteroscedasticity (GARCH) model has been often employed to improve the modelling accuracy of the AR-type model by eliminating the ARCH effect in the residuals (Fathian et al., 2019; Fathian and Vaheddoost, 2021; Pandey et al., 2019; Zha et al., 2020). However, the AR-GARCH model, a typical regression model for streamflow forecasting, has a more cumbersome combined form than a single model, hindering its application. Compared with the AR-GARCH, the double autoregressive (DAR) model proposed by Ling (2007) is more concise in form, can describe the first- and second-order moments behaviour of time series simultaneously, and has been applied maturely in the economic field (Hansen, 2021; Jiang et al., 2020; Li et al., 2019; Liu et al., 2018). In addition, the AR model requires the autoregressive and moving average parameters to fall within the range of -1~1 and the sum is lower than 1, while the GARCH model restricts non-negative and the sum (except for the constant) is below 1. In contrast, the definition range of the DAR model parameter is broader, with first-order moment selection throughout the real number field and non-negative second-order moment constraints. However, the DAR model has not yet been applied in the field of hydrology, and its prediction performance has not been verified.

Dimitriadis and Koutsoyiannis (2015) found that long-term persistence, the correlation between streamflow at the present and past moments, was very important for daily streamflow prediction. And it has been reported that time-varying volatility detected in daily streamflow series could be attributed to long-term memory (Dimitriadis et al., 2021; Graves et al., 2017; Grimaldi, 2004), which may lead to a spurious ARCH effect in residuals of the pure AR model, rendering the constructed GARCH model insufficient and affecting prediction performance. Both the combined AR-GARCH model and the novel DAR model are powerless to reproduce the long-term persistence of the daily streamflow time series. Long-term memory was always neglected in the vast majority of existing streamflow simulation and prediction studies. The research on long memory began





with Hurst (1951) and Mandelbrot and Wallis (1969), and Hurst proposed the Hurst index (H, H>0.5 indicates that the time series has long memory) to identify it. Although this characteristic was initially identified within the field of hydrology, at that time, there lacked a research methodology widely recognized by most hydrologists that could effectively reproduce the long-term memory of daily streamflow. Granger (1978) proposed the concept of "fractionally differencing", which laid the foundation for the development of later long-memory models (such as the ARFIMA model). Hosking (1981) followed up in

the field of hydrology. Montanari et al (1996; 1997) still believed that the fractional difference model is the only method to describe the daily streamflow time series with long-term persistence. Therefore, this study leads the way in hydrology by combining fractional differencing and time-varying fluctuation models to ensure the authenticity of the ARCH effect and reduce the risk of negatively affecting streamflow prediction accuracy due to insufficient information description.

Hydrological time series are highly nonlinear, as proven in numerous studies (Delforge et al., 2022; Feng et al., 2022; Guo et

al., 2021b; Miller, 2022; Yuan et al., 2021), which poses significant challenges to traditional mean models (e.g., AR, ARMA). The threshold autoregressive model based on the AR model can capture and predict the nonlinearity in the mean behaviour of hydrological time series (Tong, 1983). Several recent studies (Fathian et al., 2019; Gharehbaghi et al., 2022; Huang et al., 2021; Kolte et al., 2023) have introduced a new combination model coupled with the nonlinear mean model, namely artificial intelligence-GARCH, for prediction work. Guo et al. (2021b) combined the Self-Exciting TAR model with the GARCH model

to predict groundwater depth and achieved satisfactory prediction accuracy. However, the applicability of the TAR-GARCH model to hydrological time series is limited by narrow parameter constraints. To address this limitation, this study proposes a dual-threshold DAR (DTDAR) model based on the novel DAR model, which provides thresholds for both the linear form of the first-order and the second-order moments, namely "dual-threshold", making it more versatile than the TAR-GARCH model. As our best knowledge, the DTDAR model is proposed for the first time in this study to predict daily streamflow, and we

expect this model to become a strong competitor for time series simulation and prediction beyond streamflow.

The assumption that the residual series obeys a Gaussian distribution is imposed in traditional time series analysis methods, while actual hydrological data series have tail states that are heavier or lighter than the Gaussian distribution. Therefore, the t-distribution shape is also considered in the modelling process. This study aims to improve the prediction accuracy of daily streamflow time series by constructing a novel model with high applicability that can simultaneously capture seasonality, non-

stationarity, long-term memory, nonlinearity, and time-varying volatility of daily streamflow. Therefore, a FDTDAR model based on long memory-threshold structure is constructed based on the introduced DAR model, and seasonal normalization is used to describe seasonal effects. To test and evaluate the prediction accuracy of the DTDAR model, not only a single DAR model was compared, but also the classic AR-GARCH and TAR-GARCH models were selected.



## 2 Study area and data

### 2.1 Study area


The research area of this study (Fig. 1) is the Yellow River Basin in China, which has a length of approximately 5464 kilometres, ranking second in China and fifth in the world. The Yellow River originates from the Bayan Har Mountains in Qinghai Province, Western China, and flows through 9 provinces before emptying into the Bohai Sea. This study focuses on 15 hydrological stations within the Yellow River Basin that have been selected for their high quality and reliability. These stations

(Fig. 1) include those located on the mainstream (Tangnaihai (TNH), Lanzhou (LZ), Xiaheyan (XHY), Shizuishan (SZS), Toudaoguai (TDG), Fugu (FG), Longmen (LM), Tongguan (TG), Sanmenxia (SMX), Aishan (AS) and Lijin (LJ)) and the largest tributary Weihe River Basin (Zhangjiashan (ZJS), Xianyang (XY), Lintong (LT) and Zhuangtou (ZT)).

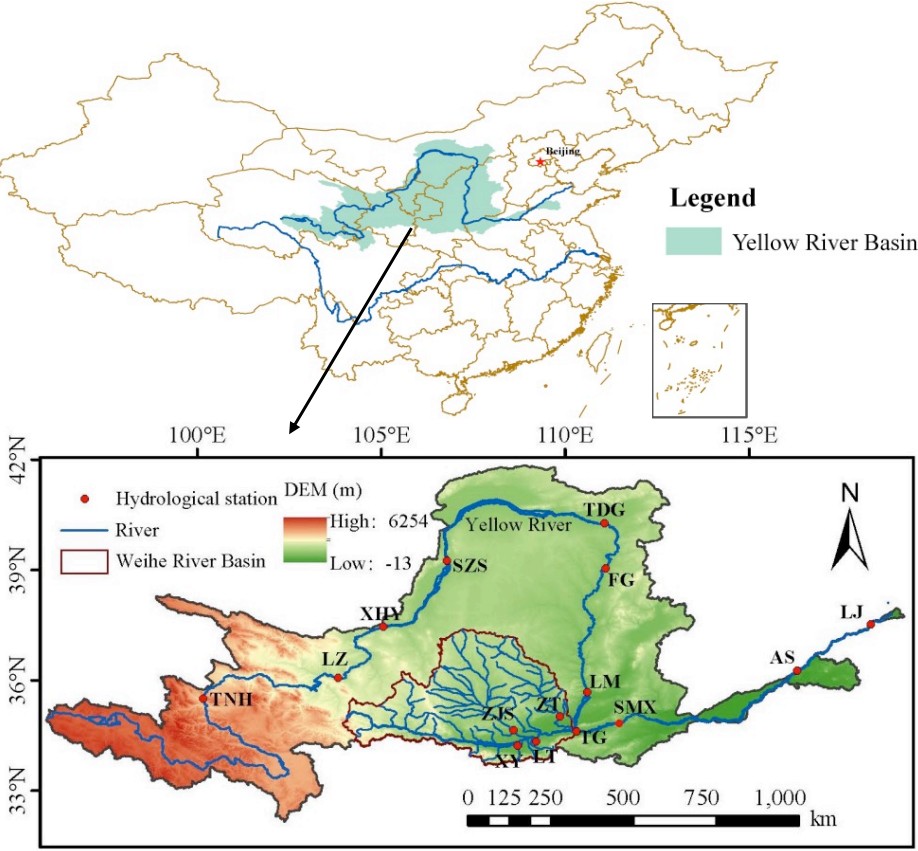

**Figure 1: The Yellow River basin and locations of 15 hydrological stations.**

**2.2 Daily streamflow time series data**

This study selects the measured daily streamflow time series from 15 hydrological stations in the Yellow River Basin (Table 1). Figure 2 depicts the temporal variation of the daily streamflow time series for the 15 hydrological stations. The last year's



data of each station is used as the testing period to compare and evaluate the prediction accuracy of various models, and the remaining serves as the training period to identify multiple characteristics of daily runoff time series, establish models, and

assess their modelling capabilities.

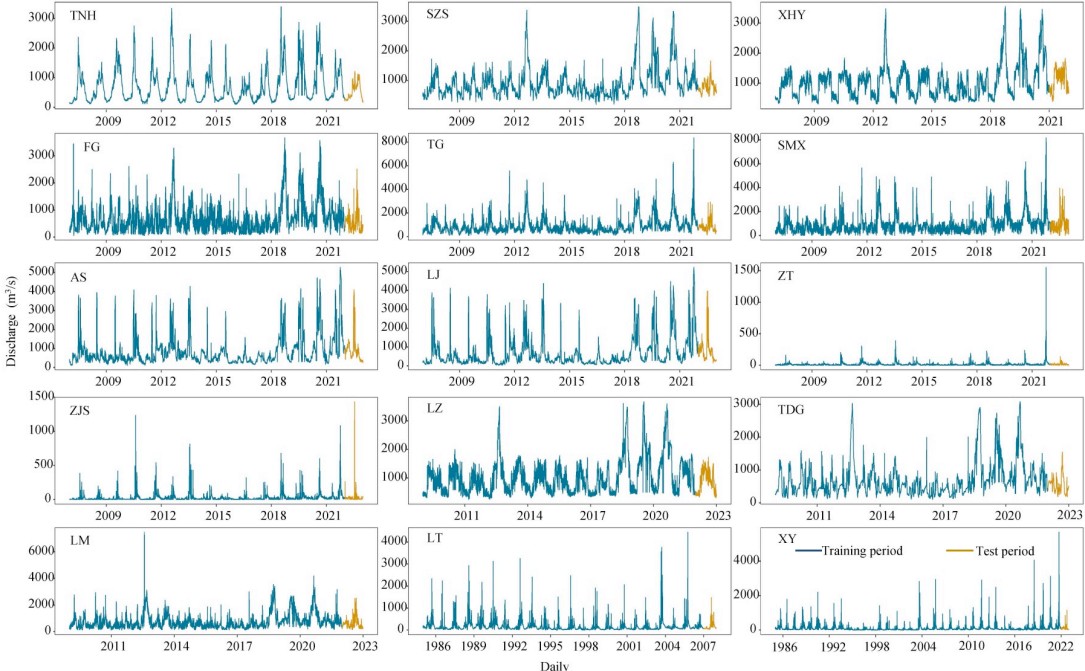

**Figure 2: Changes in daily streamflow series at 15 hydrological stations in the study area.**

**Table 1: Basic Statistics of daily streamflow series for 15 stations.**

| Stations | Abbreviation | Water system | Latitude/N | Longitude/E | Control area/km² | Period |
|---|---|---|---|---|---|---|
| Tangnaihai | TNH | Yellow River | 35.5 | 100.15 | 121972 | 2007.01.01-2022.12.31 |
| Lanzhou | LZ | Yellow River | 36.06 | 103.82 | 222600 | 2009.01.01-2022.12.31 |
| Xiaheyan | XHY | Yellow River | 37.45 | 105.05 | 254142 | 2007.01.01-2021.12.31 |
| Shizuishan | SZS | Yellow River | 39.25 | 106.78 | 309146 | 2007.01.01-2022.12.31 |



| | | | | | | |
|---|---|---|---|---|---|---|
| Toudaoguai | TDG | Yellow River | 40.27 | 111.06 | 367898 | 2009.01.01-<br>2022.12.31 |
| Fugu | FG | Yellow River | 39.03 | 111.08 | 404000 | 2007.01.01-<br>2022.12.31 |
| Longmen | LM | Yellow River | 35.67 | 110.58 | 497552 | 2009.01.01-<br>2022.12.31 |
| Tongguan | TG | Yellow River | 34.6 | 110.3 | 682141 | 2007.01.01-<br>2022.12.31 |
| Sanmenxia | SMX | Yellow River | 34.82 | 111.37 | 688421 | 2007.01.01-<br>2022.12.31 |
| Aishan | AS | Yellow River | 36.25 | 116.3 | 749136 | 2007.01.01-<br>2022.12.31 |
| Lijin | LJ | Yellow River | 37.52 | 118.3 | 751869 | 2007.01.01-<br>2022.12.31 |
| Zhangjiashan | ZJS | Jinghe | 34.64 | 108.59 | 43216 | 2007.01.01-<br>2022.12.31 |
| Xianyang | XY | Weihe River | 34.32 | 108.7 | 46827 | 1985.01.01-<br>2022.12.31 |
| Lintong | LT | Weihe River | 34.43 | 109.2 | 97299 | 1985.01.01-<br>2007.12.31 |
| Zhuangtou | ZT | Beiluohe | 35 | 109.84 | 25154 | 2007.01.01-<br>2022.12.31 |

## 2.3 Daily streamflow time series characteristics and their linkage relationships

Autocorrelation in time series refers to the correlation between a time series and a lagged version of itself. It measures the similarity or correlation between observations at different time points within the same time series. Traditional stationary time



series models (such as AR models) often depict an autocorrelation function (ACF) that exhibits a geometric series decay. In contrast, the ACF represented by long-memory models is characterized by a very slow decay and a certain persistence, clearly different from traditional models.

Figure 3 shows that the long-term memory characteristics of the daily streamflow time series are more obvious after de-seasonal standardization, with a slower decay rate and stronger persistence of ACF. Additionally, the Hurst index (H) values of the daily streamflow time series of 15 hydrological stations all exceeded 0.5, indicating the existence of long-term memory. And the H value of the original series ranged from 0.69 to 0.92, which was lower than the deseasonalization treatment (0.81 to 0.96). This suggests that seasonal factors weaken the strength of long-term memory in the daily streamflow time series,

further impacting the effectiveness of traditional models. Therefore, seasonal factors were given priority in subsequent modelling work.

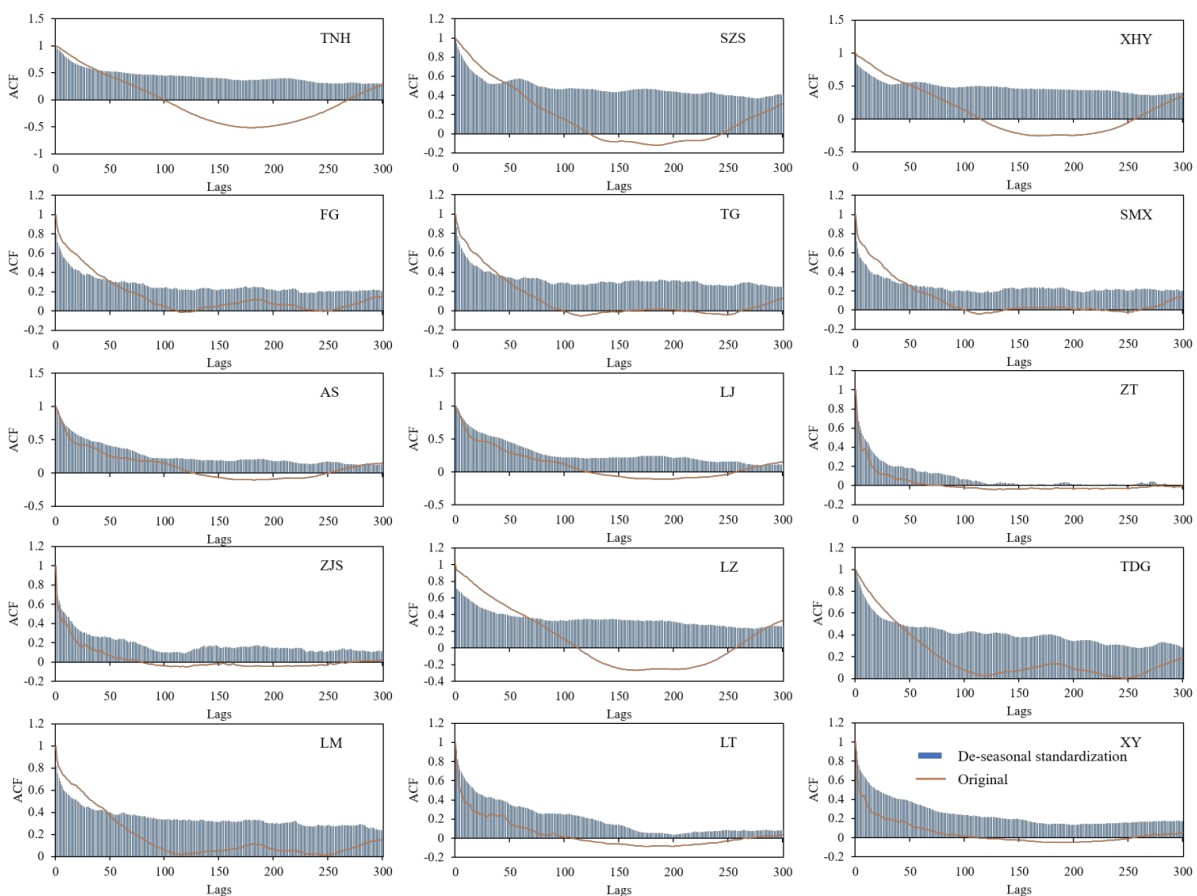

**Figure 3: Autocorrelation coefficient (ACF) of daily streamflow time series.**

Table 2 presents the results of 3 unit root tests (Augmented Dickey–Fuller (ADF), Phillips–Perron (PP), and Kwiatkowski–

Phillips–Schmidt–Shin (KPSS) tests) for the daily streamflow time series. The null hypotheses of ADF and PP tests assume that the time series has a unit root, while the KPSS test considers the null hypothesis as the time series being a stationary



process. Due to the tendency of unit root tests to favor the null hypothesis, to rigorously determine the non-stationarity of the daily runoff time series, if any of the three methods indicates the presence of a unit root process, then the daily streamflow time series at that station is considered non-stationary. The results in the table indicate that both the original and de-seasonal standardization daily streamflow time series are non-stationary processes, while after characterizing long-term memory using fractional differencing, they become stationary series.

**Table 2: p value of unit root test for daily streamflow time series**

| Stations | Original | | | De-seasonal standardization | | | Fractional difference | | |
|---|---|---|---|---|---|---|---|---|---|
| | ADF | PP | KPSS | ADF | PP | KPSS | ADF | PP | KPSS |
| TNH | >0.1 | <0.05 | >0.05 | <0.05 | <0.01 | >0.05 | <0.01 | <0.01 | >0.1 |
| SZS | >0.1 | <0.01 | <0.05 | >0.05 | <0.01 | <0.05 | <0.01 | <0.01 | >0.1 |
| XHY | >0.1 | <0.01 | <0.05 | >0.1 | <0.01 | <0.05 | <0.01 | <0.01 | >0.1 |
| FG | >0.1 | <0.01 | <0.05 | >0.05 | <0.01 | <0.05 | <0.01 | <0.01 | >0.1 |
| TG | >0.1 | <0.01 | <0.05 | >0.1 | <0.01 | <0.05 | <0.01 | <0.01 | >0.1 |
| SMX | >0.1 | <0.01 | <0.01 | >0.1 | <0.01 | <0.01 | <0.01 | <0.01 | >0.1 |
| AS | >0.1 | <0.01 | <0.05 | >0.1 | <0.01 | <0.05 | <0.01 | <0.01 | >0.1 |
| LJ | >0.1 | <0.01 | <0.05 | >0.1 | <0.01 | <0.05 | <0.01 | <0.01 | >0.1 |
| ZT | >0.1 | <0.01 | >0.1 | <0.01 | <0.01 | >0.1 | <0.01 | <0.01 | >0.1 |
| ZJS | >0.1 | <0.01 | <0.05 | >0.05 | <0.01 | <0.01 | <0.01 | <0.01 | >0.1 |
| LZ | >0.1 | <0.01 | <0.05 | >0.05 | <0.01 | <0.05 | <0.01 | <0.01 | >0.1 |
| TDG | >0.1 | <0.01 | >0.05 | >0.05 | <0.01 | <0.05 | <0.01 | <0.01 | >0.1 |
| LM | >0.1 | <0.01 | >0.05 | >0.05 | <0.01 | <0.05 | <0.01 | <0.01 | >0.1 |
| LT | >0.05 | <0.01 | >0.05 | <0.01 | <0.01 | <0.05 | <0.01 | <0.01 | >0.1 |
| XY | >0.1 | <0.01 | >0.05 | <0.05 | <0.01 | <0.05 | <0.01 | <0.01 | >0.1 |

For a linear time series model, ARCH effects represent implicit information in the squared terms of the residual series, typically tested through autocorrelation. Therefore, the integer differencing AR model and long-memory AR (FAR) model based on fractional differencing were constructed to capture the mean information of the daily streamflow time series. Figure 4 displays the results of the Ljung-Box (LB) tests for the residuals and their squared terms for the two linear models. Within the lag range of 1 ~ 20, the p-values obtained from the LB tests for the residual series of both linear models are above 0.05, satisfying the





null hypothesis of independent and uncorrelated features. This implies that the residual series of both models do not exhibit autocorrelation, indicating the effectiveness of the model results. Furthermore, a comparison reveals that the independence of

FAR model residuals is stronger, suggesting that considering long-term memory contributes to improving the ability of a model to capture information in the daily streamflow series, thereby enhancing estimation performance. The p-values for the LB tests of the squared residuals in the two models are below 0.05, indicating existing autocorrelation in the squared terms, i.e., ARCH effects. This implies that the linear models have not captured the second-moment information in the daily streamflow time series. The Lagrange Multiplier (LM) test method has been recognized as an effective means to test for ARCH effects in time

series. Calculations show that the p-values for the LM tests of the residual series for AR and FAR models are all 0, indicating a significant presence of ARCH effects in the daily streamflow time series, which is consistent with the conclusions from the LB tests mentioned above.

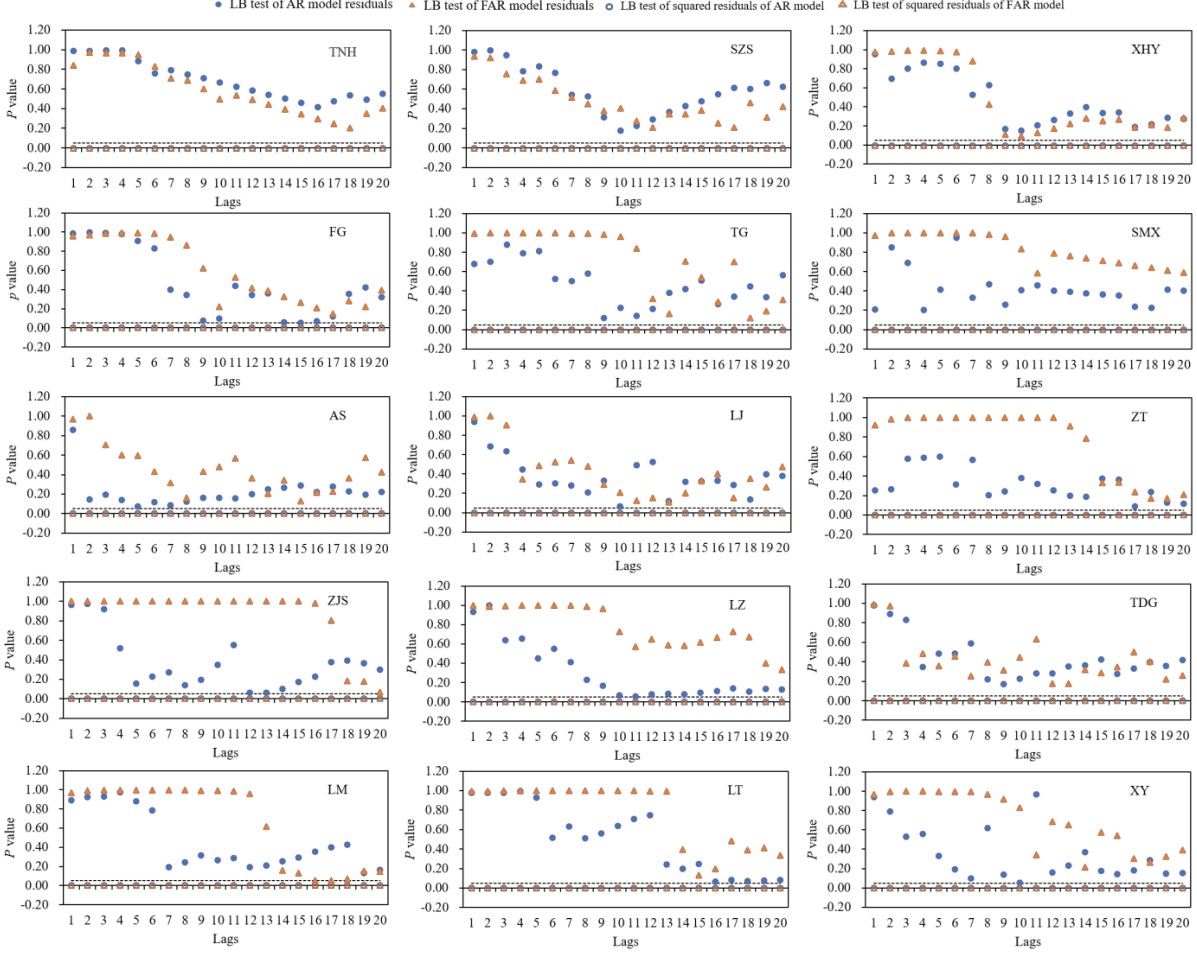

**Figure 4: Residual test of mean model of 15 hydrological stations.**





The BDS test method was employed to identify the nonlinearity in the daily streamflow time series at 15 hydrological stations, considering three different scenarios: original, de-seasonal standardization, and fractional differencing (Table 3). The range of statistic values for the BDS test of the daily streamflow time series varies significantly under different treatments, with the fractional differencing treatment yielding statistic values consistently below 40, notably lower than both de-seasonal standardization and original daily streamflow. However, the BDS test p-values are all below 0.05, indicating the presence of

nonlinear characteristics in the daily streamflow time series for original, de-seasonal standardization, and fractional differencing.

**Table 3: BDS test of daily streamflow time series**

| Stations | Original | | De-seasonal standardization | | Fractional difference | |
|---|---|---|---|---|---|---|
| | Statistics | $p$ | Statistics | $p$ | Statistics | $p$ |
| TNH | 256.41 | 0.00 | 873.66 | 0.00 | 26.80 | 0.00 |
| SZS | 212.55 | 0.00 | 283.26 | 0.00 | 25.27 | 0.00 |
| XHY | 456.70 | 0.00 | 172.65 | 0.00 | 35.21 | 0.00 |
| FG | 149.94 | 0.00 | 154.59 | 0.00 | 19.96 | 0.00 |
| TG | 131.41 | 0.00 | 198.47 | 0.00 | 20.56 | 0.00 |
| SMX | 124.00 | 0.00 | 145.50 | 0.00 | 21.73 | 0.00 |
| AS | 141.12 | 0.00 | 234.00 | 0.00 | 32.52 | 0.00 |
| LJ | 126.58 | 0.00 | 230.73 | 0.00 | 34.36 | 0.00 |
| ZT | 80.28 | 0.00 | 98.27 | 0.00 | 32.38 | 0.00 |
| ZJS | 78.64 | 0.00 | 79.35 | 0.00 | 38.85 | 0.00 |
| LZ | 235.14 | 0.00 | 128.47 | 0.00 | 34.31 | 0.00 |
| TDG | 189.29 | 0.00 | 303.57 | 0.00 | 28.70 | 0.00 |
| LM | 128.11 | 0.00 | 162.02 | 0.00 | 22.23 | 0.00 |
| LT | 132.69 | 0.00 | 241.98 | 0.00 | 50.77 | 0.00 |
| XY | 153.28 | 0.00 | 288.36 | 0.00 | 59.23 | 0.00 |



## 3 Methodology

### 3.1 De-seasonal standardization

For a given original daily streamflow time series, the mathematical expression for deseasonalization is as follows (Guo et al., 2021a):

$$\begin{cases} \mu^m = \frac{1}{N}\sum_{n=1}^{N} X_t^{(n,m)} \\ \sigma^m = \left[\frac{1}{N}\sum_{n=1}^{N}\left(X_t^{(n,m)} - \mu^m\right)^2\right]^{0.5} \end{cases} \tag{1}$$

$$Y_t = \frac{X_t - \mu^m}{\sigma^m} \tag{2}$$

where $X_t^{(n,m)}$ $(n = 1, \cdots, N; m = 1, \cdots, M)$ ($M$=366 in a leap year and 365 in a normal year) is the original daily streamflow

time series of the m-th day in the n-th year, $\mu^m$ and $\sigma^m$ are the seasonal mean and variance respectively; $Y_t$ $(t = 1, \cdots, T)$ represents the daily streamflow time series after removing seasonality, and the training period length is $T$.

### 3.2 FDTDAR model

The fractional difference is used to describe the long-term memory of the daily streamflow series ($Y_t$) after removing the seasonal components. The specific mathematical form is expressed as:

$$y_t = (1 - L)^d Y_t \tag{3}$$

where $y_t$ is the daily streamflow series after difference, $d$ is the difference order, and $LY_t = Y_{t-1}$ is the lag operator. In order to facilitate the calculation, the binomial expansion of $(1 - L)^d$ is performed as follows:

$$(1 - L)^d = 1 - dL + \frac{d(d-1)L^2}{2!} - \frac{d(d-1)(d-2)L^3}{3!} + \cdots\cdots \tag{4}$$

Eq. (4) shows that infinite data information before the observation point needs to be used in the difference process, which is

almost impossible in actual calculation, and a reasonable approximation is necessary. Therefore, Eq. (4) is substituted into Eq. (3) and sorted to obtain the difference formula used in this study. The specific structure is as follows:

$$y_t = \sum_{k=0}^{t} \omega_k \cdot Y_{t-k} \tag{5}$$

$$\omega_k = -\omega_{k-1}\frac{d-k+1}{k} \tag{6}$$

where the weight $\omega_k$ ($\omega_0 = 1$) means that the amount of information required for each data point is different, and $k$ represents

the time lag.

The long-memory DAR (FDAR) combines linearity and nonlinearity of time series, and its general structure is as follows (Ling, 2007):

$$y_t = \varphi + \sum_{i=1}^{p} a_i y_{t-i} + \varepsilon_t \sqrt{\alpha + \sum_{i=1}^{p} b_i y_{t-i}^2} \tag{7}$$





where $y_t$ is the daily streamflow time series processed by fractional difference (Eq. (5)). $\varphi$ and $\alpha$ are constants, $a_1 \cdots a_p$ and

$b_1 \cdots b_p$ are the coefficients of the FDAR model with the order $p$, $\varepsilon_t$ represents the residual series with mean 0 and variance 1.

Eq. (7) shows that the mathematical form used to describe the first-order and second-order moment characteristics of the time

series in FDAR model are linear structures, that is $A = \varphi + \sum_{i=1}^{p} a_i y_{t-i}$ and $B = \omega + \sum_{i=1}^{p} b_i y_{t-i}^2$. Thus, both parts are

assigned thresholds $r_1$ and $r_2$ to build the FDTDAR model, and the corresponding lag orders are $d_1$ and $d_2$ respectively. The

$\Theta$ is the set of model parameters. When $d_1 \neq d_2$, the mathematical form of the FDTDAR model is expressed as:

$$
\quad y_t = \begin{cases}
\varphi_{11} + \sum_{i=1}^{p_{11}} a_{1i}^1 y_{t-i} + \varepsilon_t \sqrt{\alpha_{11} + \sum_{i=1}^{q_{11}} b_{1i}^1 y_{t-i}^2} & if \ y_{t-d_1} \leq r_1 \ \ y_{t-d_2}^2 \leq r_2 \\
\varphi_{21} + \sum_{i=1}^{p_{21}} a_{1i}^2 y_{t-i} + \varepsilon_t \sqrt{\alpha_{21} + \sum_{i=1}^{q_{12}} b_{1i}^2 y_{t-i}^2} & if \ y_{t-d_1} > r_1 \ \ y_{t-d_2}^2 \leq r_2 \\
\varphi_{12} + \sum_{i=1}^{p_{12}} a_{2i}^1 y_{t-i} + \varepsilon_t \sqrt{\alpha_{12} + \sum_{i=1}^{q_{21}} b_{2i}^1 y_{t-i}^2} & if \ y_{t-d_1} \leq r_1 \ \ y_{t-d_2}^2 > r_2 \\
\varphi_{22} + \sum_{i=1}^{p_{22}} a_{2i}^2 y_{t-i} + \varepsilon_t \sqrt{\alpha_{22} + \sum_{i=1}^{q_{22}} b_{2i}^2 y_{t-i}^2} & if \ y_{t-d_1} > r_1 \ \ y_{t-d_2}^2 > r_2
\end{cases} \tag{8}
$$

where $\varphi_{11}, \varphi_{21}, \varphi_{12}, \varphi_{22}, \alpha_{11}, \alpha_{21}, \alpha_{12}$ and $\alpha_{22}$ are constants, represented by the set $C$; the order of the model is $p_{11}, p_{21}$,

$p_{12}, p_{22}, q_{11}, q_{12}, q_{21}$, and $q_{22}$, and the set is $O$; $a_{11}^1 \cdots a_{1p_{11}}^1, a_{11}^2 \cdots a_{1p_{21}}^2, a_{21}^1 \cdots a_{2p_{12}}^1, a_{21}^2 \cdots a_{2p_{22}}^2$ and $b_{11}^1 \cdots b_{1q_{11}}^1, b_{11}^2 \cdots$

$b_{1q_{12}}^2, b_{21}^1 \cdots b_{2q_{21}}^1, b_{21}^2 \cdots b_{2q_{22}}^2$ represent the first-order and second-order moment coefficients, respectively, which expressed

by the set E. When $d_1 = d_2 = d_0$, the threshold of FDTDAR ($\Theta$) model is $r_1 = r_2 = r_0$, and its general form is expressed as:

$$
\quad y_t = \begin{cases}
\varphi_{10} + \sum_{i=1}^{p_1} a_{1i} y_{t-i} + \varepsilon_t \sqrt{\alpha_{10} + \sum_{i=1}^{q_1} b_{1i} y_{t-i}^2} & if \ y_{t-d_0} \leq r_0 \\
\varphi_{20} + \sum_{i=1}^{p_2} a_{2i} y_{t-i} + \varepsilon_t \sqrt{\alpha_{20} + \sum_{i=1}^{q_2} b_{2i} y_{t-i}^2} & if \ y_{t-d_0} > r_0
\end{cases} \tag{9}
$$

where $\varphi_{10}, \varphi_{20}, \alpha_{10}$ and $\alpha_{20}$ are constants, $a_{11} \cdots a_{1p_1}, a_{21} \cdots a_{2p_2}, b_{11} \cdots b_{1q_1}$, and $b_{21} \cdots b_{2q_2}$ represent the coefficients of

the model with orders $p_1, p_2, q_1$ and $q_2$.

In fact, Li et al. (2016) proposed a similar form of Eq. (9) ($d_1 = d_2$) and further studied the quasi-maximum likelihood

estimation of the model. The significant difference between this model and the TAR-GARCH model is that the former makes

the dynamic behavior of the conditional variance visible by specifying it in the observation function. Daren and Huay-min

(2004) investigated the structure of such models in general settings such as the strict stationarity and V-uniform ergodicity.

The FDTDAR model we proposed (Eq. (8)) is further extended based on the work of previous researchers. For a further

discussion on the structure of this model, please refer to Li et al. (2016) and Daren and Huay-min (2004).

### 3.3 Parameter estimation of FDTDAR model

(1) Gaussian distribution

Assume that the time series $y_t$ is a sample from the FDTDAR model (taking Eq. (8) as an example), given an initial value

$\{y_{t-p}, \cdots, y_0\}$, where $p = max\{p_{11}, p_{21}, p_{12}, p_{22}, q_{11}, q_{12}, q_{21}, q_{22}\}$. The conditional log-likelihood function is defined as:





$$Ln(\theta) = \sum_{t=1}^{T} l_t(\theta) = \sum_{t=1}^{T} \left[ -\frac{1}{2} log h_t(\theta) - \frac{[y_t - u_t(\theta)]^2}{2h_t(\theta)} \right] \tag{10}$$

where $\theta$ is the set of model parameters, $\theta = (C', E', r_1, r_2)' \equiv (C', A', B', r_1, r_2)'$. And $C' = (\varphi_{11}, \varphi_{21}, \varphi_{12}, \varphi_{22}, \alpha_{11}, \alpha_{21}, \alpha_{12}, \alpha_{22})'$, $A = (A_1, A_2, A_3, A_4)$, $A_1 = \left( a_{11}^1 \cdots a_{1p_{11}}^1 \right)$, $A_2 = \left( a_{11}^2 \cdots a_{1p_{21}}^2 \right)$, $A_3 = \left( a_{21}^1 \cdots a_{2p_{12}}^1 \right)$, $A_4 = \left( a_{21}^2 \cdots a_{2p_{22}}^2 \right)$; $B_1 = \left( b_{11}^1 \cdots b_{1q_{11}}^1 \right)$, $B_2 = \left( b_{11}^2 \cdots b_{1q_{12}}^2 \right)$, $B_3 = \left( b_{21}^1 \cdots b_{2q_{21}}^1 \right)$, $B_4 = \left( b_{21}^2 \cdots b_{2q_{22}}^2 \right)$. $u_t(\theta)$ and $h_t(\theta)$ represent the conditional mean and conditional variance of the FDTDAR model respectively. Their specific mathematical expressions are:

$$u_t(\theta) = \left( \varphi_{11} + A_1' Z_{1,t-1} \right) I\left( y_{t-d_1} \le r_1, y_{t-d_2}^2 \le r_2 \right)$$
$$+ \left( \varphi_{21} + A_2' Z_{2,t-1} \right) I\left( y_{t-d_1} > r_1, y_{t-d_2}^2 \le r_2 \right)$$
$$+ \left( \varphi_{12} + A_3' Z_{3,t-1} \right) I\left( y_{t-d_1} \le r_1, y_{t-d_2}^2 > r_2 \right)$$
$$+ \left( \varphi_{22} + A_4' Z_{4,t-1} \right) I\left( y_{t-d_1} > r_1, y_{t-d_2}^2 > r_2 \right)$$

(11)

$$h_t(\theta) = \left( \alpha_{11} + B_1' S_{1,t-1} \right) I\left( y_{t-d_1} \le r_1, y_{t-d_2}^2 \le r_2 \right)$$
$$+ \left( \alpha_{21} + B_2' S_{2,t-1} \right) I\left( y_{t-d_1} > r_1, y_{t-d_2}^2 \le r_2 \right)$$
$$+ \left( \alpha_{12} + B_3' S_{3,t-1} \right) I\left( y_{t-d_1} \le r_1, y_{t-d_2}^2 > r_2 \right)$$
$$+ \left( \alpha_{22} + B_4' S_{4,t-1} \right) I\left( y_{t-d_1} > r_1, y_{t-d_2}^2 > r_2 \right)$$

(12)

Where $Z_{1,t-1} = \left( 1, y_{t-1}, \cdots, y_{t-p_{11}} \right)'$, $Z_{2,t-1} = \left( 1, y_{t-1}, \cdots, y_{t-p_{21}} \right)'$, $Z_{3,t-1} = \left( 1, y_{t-1}, \cdots, y_{t-p_{12}} \right)'$, $Z_{4,t-1} = \left( 1, y_{t-1}, \cdots, y_{t-p_{22}} \right)'$; $S_{1,t-1} = \left( 1, y_{t-1}^2, \cdots, y_{t-q_{11}}^2 \right)'$, $S_{2,t-1} = (1, y_{t-1}^2, \cdots, y_{t-q_{21}}^2 )'$, $S_{3,t-1} = \left( 1, y_{t-1}^2, \cdots, y_{t-q_{12}}^2 \right)'$, $S_{4,t-1} = \left( 1, y_{t-1}^2, \cdots, y_{t-q_{22}}^2 \right)'$. $I(\cdot)$ is an indicator function.

Given the initial value of the observation, the quasi-maximum likelihood estimate (QMLE) of the parameter $\theta$ can be defined by the following formula:

$$\hat{\theta} = argmax Ln(\theta) \tag{13}$$

where $\hat{\theta}$ is the quasi-maximum likelihood estimator of parameter vector $\theta$.

(2) Student's t distribution

The normal distribution is the most commonly used conditional distribution form in the DAR model. However, for the significant peak-tailed characteristics in hydrological time series, the heavy-tailed distribution of linear model residuals may be insufficient, and even the residual series obtained from volatility are required to be heavy-tailed. Therefore, the normal distribution assumption of the classic DAR model may be difficult to meet the needs of hydrological time series modeling. It is worthwhile to introduce skewed conditional distribution into the DAR model to enhance its ability to characterize the heavy-tail characteristics of time series.





The expansion of the conditional distribution in the classic ARCH model starts from the Student's t distribution. Regarding the ARCH model, Engle (1982, 1987, 1990, 1993) mentioned after extensive research that the conditional distribution of the volatility model can adopt a non-Gaussian form. Therefore, Bollerslev (1987) first used the student's t distribution to describe the distribution form of the residuals. This study used the student's t distribution to expand the conditional distribution of the DAR model.

Therefore, the student's t distribution is considered in the FDTDAR model, that is $\varepsilon_t \xrightarrow{i.i.d} t(0,1;k)$, where k is the degree of freedom of the student's t distribution. The most common probability density function of the student's t distribution is expressed as:

$$f(y_t, k) = \frac{\Gamma\left(\frac{k+1}{2}\right)}{\sqrt{k\pi}\Gamma\left(\frac{k}{2}\right)}\left(1 + \frac{y_t^2}{k}\right)^{\frac{-(k+1)}{2}} \tag{14}$$

where $\Gamma(\cdot)$ is the Gamma function. $M_t$ is used to represent the scale parameter of the FDTDAR model, then the probability density function of the time series $y_t$ is expressed as:

$$f(y_t|M_t) = \frac{\Gamma\left(\frac{k+1}{2}\right)}{\sqrt{k\pi}\Gamma\left(\frac{k}{2}\right)}\frac{1}{\sqrt{M_t}}\left[1 + \frac{(y_t-u_t)^2}{M_t k}\right]^{\frac{-(k+1)}{2}} \tag{15}$$

When the degrees of freedom k>2, the mean of the residual series $\varepsilon_t$ is 0, and the variance $h_t$ is expressed as:

$$E(y_t^2) = \frac{M_t k}{k-2} \tag{16}$$

Therefore, for series $y_t$ with variance $h_t$ and freedom $k$, the scale parameter $M_t$ can be written as:

$$M_t = \frac{h_t(k-2)}{k} \tag{17}$$

The probability density function then becomes:

$$f(y_t|h_t) = \frac{\Gamma\left(\frac{k+1}{2}\right)}{\sqrt{(k-2)\pi}\Gamma\left(\frac{k}{2}\right)}\frac{1}{\sqrt{h_t}}\left[1 + \frac{(y_t-u_t)^2}{h_t(k-2)}\right]^{\frac{-(k+1)}{2}} \tag{18}$$

where the freedom degree $k$ is to be estimated, the conditional log-likelihood function is:

$$Ln(\theta) = \sum_{t=1}^{T} l_t(\theta) = \sum_{t=1}^{T} ln[f(y_t|h_t)]$$

$$= \sum_{t=1}^{T} ln\left[\frac{\Gamma\left(\frac{k+1}{2}\right)}{\sqrt{(k-2)\pi}\Gamma\left(\frac{k}{2}\right)}\frac{1}{\sqrt{h_t}}\left[1 + \frac{(y_t-u_t)^2}{h_t(k-2)}\right]^{\frac{-(k+1)}{2}}\right] \tag{19}$$

$$= -\frac{T}{2}ln\left[\frac{\pi(k-2)\Gamma\left(\frac{k}{2}\right)^2}{\Gamma\left(\frac{k+1}{2}\right)^2}\right] - \frac{1}{2}\sum_{t=1}^{T} lnh_t - \frac{k+1}{2}\sum_{t=1}^{T} ln\left[1 + \frac{(y_t-u_t)^2}{h_t(k-2)}\right]$$

where $u_t$ and $h_t$ are calculated by Eq. (11) and Eq. (12) respectively. If there exists a vector $\hat{\theta}$ such that $Ln(\theta)$ has the maximum value, then in the setting of maximum likelihood estimation, vector $\hat{\theta}$ is considered as the maximum likelihood estimator for the parameter vector $\theta$, and it is determined by Eq. (13).





### 3.4 Order determined of FDTDAR model

Obviously, as the order increases, the FDTDAR model's ability to describe time series becomes stronger. However, the accompanying large parameter set adds complexity to the model structure and reduces computational speed. Therefore, it is

necessary to use a metric tool to determine the optimal order for the model. The evaluation of model quality generally depends on two aspects: the likelihood function value of parameter estimation and the number of unknown parameters in the model. A larger likelihood function value and a greater number of parameters indicate superior model fitting. However, the risk of "overfitting" arises when the simulation performance is excessively superior, leading to a decrease in accuracy during the prediction phase.

The process of model order determination is essentially an optimization task aimed at balancing the two aspects mentioned above. In practical computations, we aim for a larger likelihood function value while minimizing the number of model parameters. The Akaike Information Criterion (AIC) exhibits outstanding effectiveness in the context of model order determination. The calculation formula of AIC based on DTDAR model is:

$$AIC(p) = -2Ln(\hat{\theta}) + 2(p_{11} + p_{21} + p_{12} + p_{22} + q_{11} + q_{12} + q_{21} + q_{22} + 8) \tag{20}$$

During the order determination process, restrict the value of p within the range of 1 to 25 to select the minimum AIC value. And $p_{ij}$ $(i, j = 1,2)$ and $q_{ij}$ $(i, j = 1,2)$ take values within $[1, p]$, $d_1 \in [1, min(p_{ij})]$, $d_2 \in [1, min(q_{ij})]$, $r_1 \in [min(y_t), max(y_t)]$, $r_2 \in [min(y_t^2), max(y_t^2)]$.

### 3.5 FTAR-GARCH model

The TAR model (Tong, 1983) is based on the AR model by adding a threshold to achieve the nonlinear description of the time

series mean behavior. The mathematical structure of a two-stage TAR model is written as:

$$y_t = \begin{cases} \omega_{10} + \sum_{i=1}^{p^1} \beta_i^1 y_{t-i} + e_t^1 & if \ y_{t-c} \leq \tau \\ \omega_{20} + \sum_{i=1}^{p^2} \beta_i^2 y_{t-i} + e_t^2 & if \ y_{t-c} > \tau \end{cases} \tag{21}$$

where $\omega_{10}$ and $\omega_{20}$ are constants, $\beta_i^1$ and $\beta_i^2$ represent the coefficients of model with orders $p^1$ and $p^2$, the $e_t^1$ and $e_t^2$ are residuals with mean 0 and variance $\sigma^2$, and $c$ and $\tau$ express specified time delay and threshold, respectively. The conditional heteroscedasticity information in the residual series of the TAR model is captured by the GARCH (1,1) model (Bollerslev,

1986), and its specific form is as follows:

$$\sigma_t^2 = \omega + ve_{t-1}^2 + \theta\sigma_{t-1}^2 \tag{22}$$

$$e_t = \sigma_t \eta_t \quad \eta_t \sim N(0, 1) \tag{23}$$

where $\omega$ is a constant, $v$ and $\theta$ are coefficients of the GARCH model, $\sigma_t^2$ is the condition time-varying variance of the residual series.





### 3.6 Comparative evaluation methods

Five indicators are used in this study to compare and evaluate the prediction performance of the models, namely Mean Absolute Error (MAE), Root Mean Squared Error (RMSE), Coefficient of determination ($R^2$), Nash-Sutcliffe efficiency coefficient (NSE), and Absolute Maximum Error (AME) (Dawson et al., 2007; Liu et al., 2020; Moriasi et al., 2007). The smaller the MAE and RMSE, the closer the $R^2$ and NSE are to 1, indicating that the model prediction performance is better.

In addition, the DM test is used to compare the accuracy of two models (model 1 and model 2), and its null hypothesis is that the prediction accuracy of model 1 is higher than that of model 2.

### 4 Results

Figure 5 illustrates the presence of the FDTDAR model at various hydrological stations. Under the assumption of normal distribution, the values of $d_1$ and $d_2$ in the FDTDAR model of TG, SMX, LM, LJ, and AS hydrological stations are different, while in the Student's t distribution, they are different for TG, SMX, LM, FG, and AS stations. Additionally, in the FDTDAR models with equal lag times (Eq. (9)), the value of $d_0$ is consistently 1, while in another form (Eq. (8)), the lag times ($d_1$ and $d_2$) have values of 1 or 2. Furthermore, the order of FDTDAR-n and FDTDAR-t models generally ranges from 1 to 5 days, except for ZT and LJ stations where the FDTDAR-t model order reaches 20.

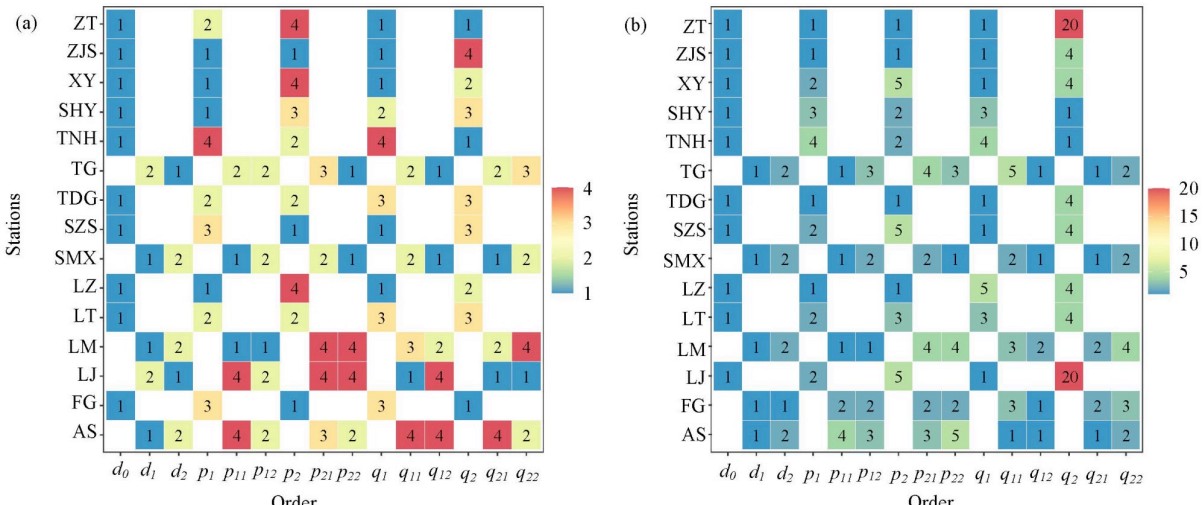

**Figure 5: The order of FDTDAR model in 15 hydrological stations, where (a) and (b) represent the normal distribution and the student's t distribution respectively.**

Overall, although the FDTDAR model based on Eq. (9) is chosen by the majority of stations, approximately one-third of the stations, mostly located in the mid to downstream areas of the basin, lean towards another pattern. Due to intensified climate change and enhanced human activities, the nonlinearity of daily streamflow time series may continue to strengthen. FDTDAR models with the same lag time essentially use a threshold at the mean level to simultaneously constrain the behavior of the first and second moments, which may be challenging in addressing the challenges brought about by nonlinear changes at the



second-moment level in daily streamflow time series. Except for ZT and LJ stations, the complexity of the FDTDAR models under two residual distribution forms is generally consistent.

The thresholds for the FDTDAR model with different residual distributions at 15 hydrological stations are presented in Table 4. The inputs for each model are daily streamflow time series that have undergone seasonal standardization and fractional differencing. The thresholds for the FDTDAR models based on two residual distributions are identical for the TNH, XHY, SMX, ZJS, LM, and LT hydrological stations.

**Table 4: Thresholds of FDTDAR model in 15 hydrological stations**

| Stations | Distributions | $r_1$ | $r_2$ | $r_0$ | Stations | Distributions | $r_1$ | $r_2$ | $r_0$ |
|---|---|---|---|---|---|---|---|---|---|
| TNH | n | | | -0.71 | ZT | n | | | 2.72 |
| | t | | | -0.71 | | t | | | -1.28 |
| SZS | n | | | -0.25 | ZJS | n | | | 1.41 |
| | t | | | 2.4 | | t | | | 1.41 |
| XHY | n | | | 1.79 | LZ | n | | | 0.79 |
| | t | | | 1.79 | | t | | | 1.79 |
| FG | n | | | -0.69 | TDG | n | | | -1.7 |
| | t | -3.69 | 7 | | | t | | | -0.7 |
| TG | n | -3.2 | 2.56 | | LM | n | 1.98 | 3 | |
| | t | 3.04 | 10.1 | | | t | 1.98 | 3 | |
| SMX | n | -3.42 | 10.7 | | LT | n | | | 1.47 |
| | t | -3.42 | 10.7 | | | t | | | 1.47 |
| AS | n | 0.52 | 3 | | XY | n | | | 1.54 |
| | t | -1.76 | 4.63 | | | t | | | 2.09 |
| LJ | n | -1.88 | 3 | | | | | | |
| | t | | | -1.88 | | | | | |

## 4.2 Comparison between FDTDAR and FTAR-GARCH models

Figure 6 compares the predictive performance of various long memory threshold models at 15 hydrological stations. For mainstream hydrological stations, higher levels of average error (MAE and MRE) and extreme error (RMSE and AME), and





lower $R^2$ and NSE values, indicate poor predictive accuracy for the four models at the SMX station. From a model category perspective, it can be observed that for most stations, the FDTDAR class models have smaller MAE, RMSE, MRE, and AME values than the FTAR-GARCH class models. Both model classes show similar predictive accuracy, with $R^2$ and NSE values above 0.85 at stations other than SMX. FDTDAR-type models are more influenced by the residual distribution shape, and the FDTDAR-t model has lower MAE, RMSE, and MRE values than the FDTDAR-n model. However, there is no significant difference in the three metrics for FAR-GARCH class models based on two distributions. In summary, FDTDAR models exhibit superior predictive ability in mainstream hydrological stations compared to FTAR-GARCH models, and the student's t distribution is more suitable than the normal distribution for describing the changing form of daily streamflow.

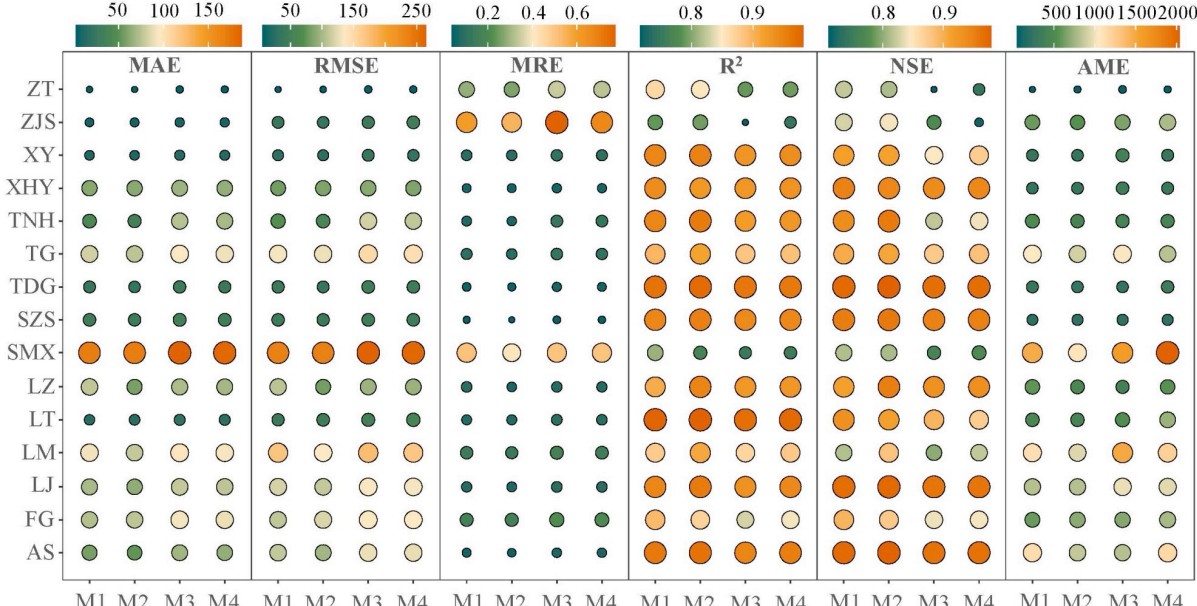

**Figure 6: Prediction evaluation indicators of long memory threshold models, where M1, M2, M3 and M4 are FDTDAR-n, FDTDAR-t, FTAR-GARCH-n and FTAR-GARCH-t models respectively.**

Furthermore, Figure 6 illustrates that, in the case of four tributary hydrological stations (ZT, ZJS, LT, and XY), the FDTDAR type models exhibit relatively lower values of MAE, RMSE, MRE, and AME, while simultaneously having higher R and NSE values compared to FTAR-GARCH models. The predictive results of FDTDAR and FTAR-GARCH models vary under different distribution assumptions, with the t-distribution yielding superior predictive performance. In short, FDTDAR models demonstrated outstanding predictive capabilities in the daily streamflow time series at tributary hydrological stations, and the t-distribution significantly improved the accuracy of model predictions.





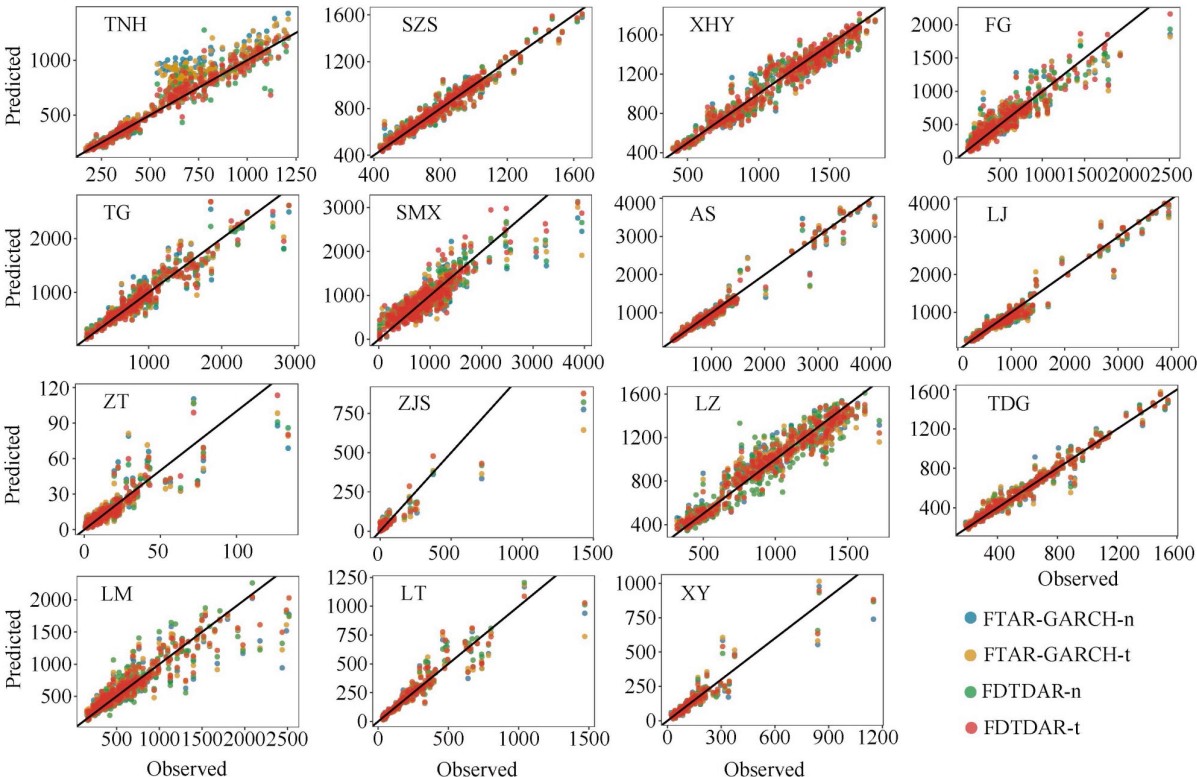

**Figure 7: Prediction scatter points of FDTDAR and FTAR-GARCH models at 15 hydrological stations.**

Predictive performance of the FDTDAR-n, FDTDAR-t, FTAR-GARCH-n, and FTAR-GARCH-t models at 15 hydrological stations reveal that, for the majority of stations, the degree of clustering of predicted points around the 1:1 line is stronger for the FDTDAR models compared to the FTAR-GARCH models (Fig. 7). Under the assumption of student's t distribution, both FDTDAR and FTAR-GARCH models exhibit better matching with observed daily streamflow compared to their counterparts assuming normal distribution (FDTDAR-n and FTAR-GARCH-n models). This suggests that the FDTDAR models have stronger predictive capabilities for daily streamflow time series compared to the FTAR-GARCH models, and the student's t distribution improves the predictive performance and peak description ability of the models under the classical normal distribution assumption.





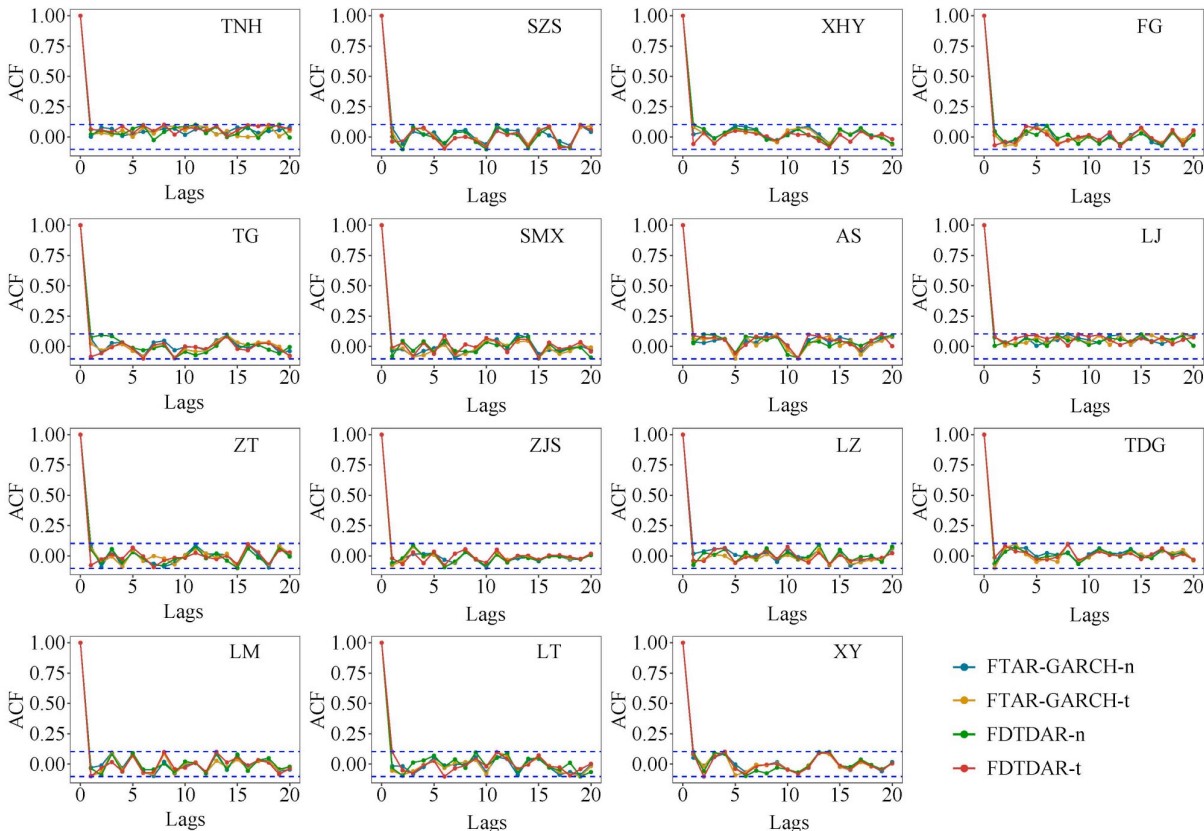

**Figure 8: Prediction scatter points of FDTDAR and FTAR-GARCH models at 15 stations.**

Figure 8 shows that the ACF of the residual series from the long memory threshold FDTDAR and FTAR-GARCH models, based on two different residual distributions, are within the confidence intervals, indicating the absence of autocorrelation in the residual series. This implies that both model types effectively capture practical information in the daily streamflow time series, and the model predictions are reliable.

**4.3 Effectiveness evaluation of long memory threshold structure**

This section compares the point and interval prediction performance of DAR-type and AR-GARCH-type models under three different structures (integer differencing, long memory, and long memory threshold) and two residual distribution assumptions. In the point prediction comparison, $R^2$ and NSE values are chosen to assess the prediction accuracy of model. And for interval prediction comparison, average interval width (AIW) and containing ratio (CR) values are selected to assess the prediction uncertainty and the inclusion rate of observed values. Finally, the DM test is employed to identify the optimal prediction model structure and type for each station.

Figure 9 reveals that, under the "long memory threshold" structure that considers nonlinear changes, both AR-GARCH-type and DAR-type models exhibit higher $R^2$ and NSE values than the two linear structures. Furthermore, the FAR-GARCH and





FDAR models, which account for long-term memory, significantly improve the NSE values of the classical AR-GARCH and
DAR models based on integer differencing. This indicates that the long-term memory and strong nonlinearity in the daily
streamflow time series significantly influence the predictive performance of the models, and the long memory threshold
structure is effective in improving the accuracy of daily streamflow predictions.





**Figure 9: Comparison of prediction accuracy of DAR-type and AR-GARCH-type models based on three structures**
**with different residual distributions.**

Figure 10 and Figure 11 show that the prediction interval width of DAR-type and AR-GARCH-type models with the "long memory-threshold" structure, considering long-term memory and nonlinear changes, is significantly narrower than those of the "long memory" and "integer differencing" linear structures. However, different settings have minimal impact on the CR values of the models. This indicates that FDTDAR and FAR-TGARCH models include more observed values within narrower
prediction intervals, and the "long memory threshold" structure significantly improves the interval prediction performance of the models compared to the single "long memory" and "integer differencing" structures.

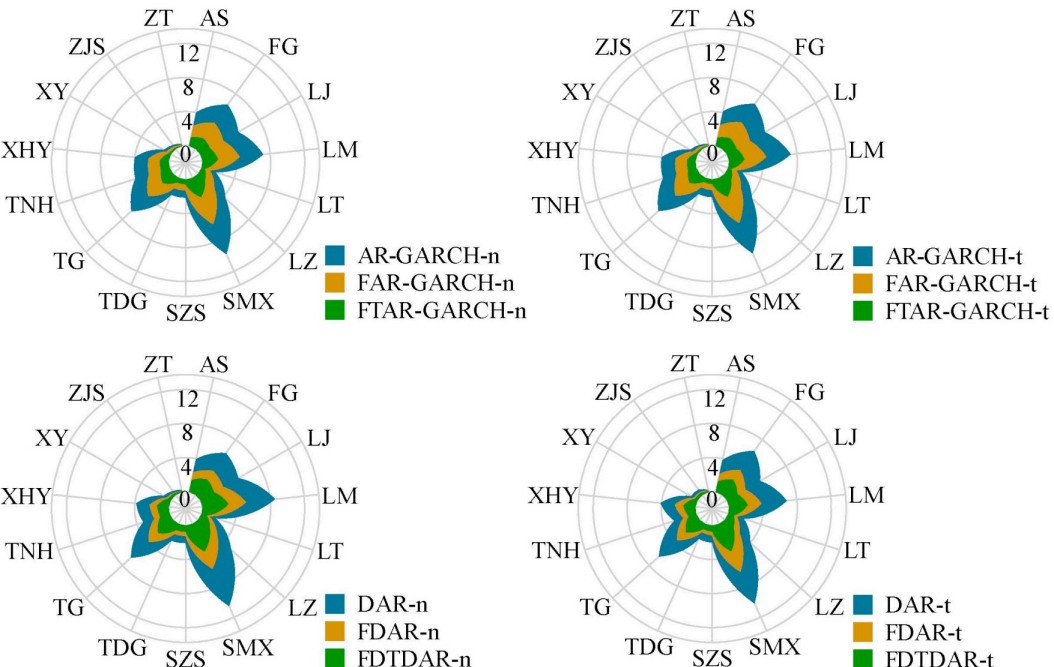

**Figure 10: Comparison of AIW of DAR and AR-GARCH models based on three structures.**



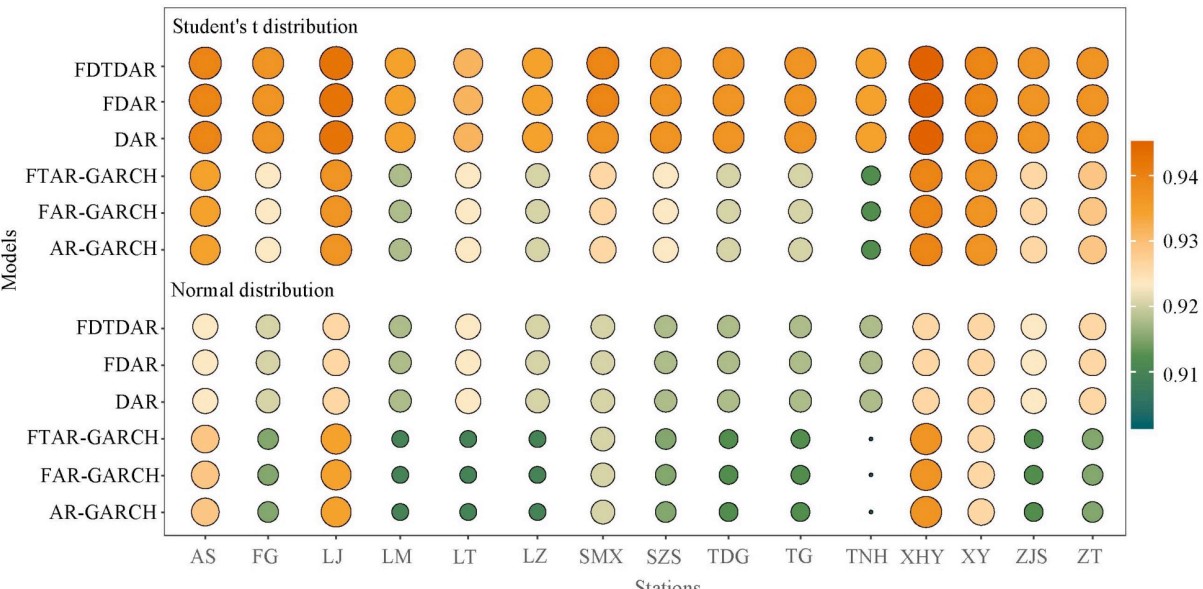

**Figure 11: Comparison of CR of DAR and AR-GARCH models based on three structures.**

The comprehensive evaluation results of the predictive performance of DAR-type and AR-GARCH-type models under different structures are presented in Table 5. In the group of Linear vs. Nonlinear, the linear models are derived from the advantageous models in the difference group, while the two superior models in the residual distribution comparison groups come from the optimal models of each distribution. It can be observed that, except for the ZJS station, the long memory structure based on fractional differencing effectively enhances the predictive ability of the two type models under traditional integer differencing. The p-values less than 0.05 indicate that the long-memory threshold structure significantly contributes to improving the prediction accuracy of the two type models (DAR and AR-GARCH) with linear changing characteristics (excluding the ZJS station). Additionally, the results of different residual comparison group indicate that TNH, SZS, LJ, LZ, TDG, and LM stations are more suitable for the t-distribution in DAR-type model, while in AR-GARCH-type, only TNH and XHY stations exhibit this preference.

**Table 5.** *p*-value of DM test for DAR-type and AR-GARCH-type models under different structures (M1 vs. M2 shows that the original hypothesis of the DM test is that the prediction performance of M1 is better than M2).

| Stations | DAR-type model | | | | | AR-GARCH-type model | | | | |
|---|---|---|---|---|---|---|---|---|---|---|
| | Normal distribution | | Student's t distribution | | | Normal distribution | | Student's t distribution | | |
| | Difference | Linear vs. | Difference | Linear vs. | n vs. t | Difference | Linear vs. | Difference | Linear vs. | n vs. t |
| | Integer vs. Fraction | Nonlinear | Integer vs. Fraction | Nonlinear | | Integer vs. Fraction | Nonlinear | Integer vs. Fraction | Nonlinear | |
| TNH | 0 | 0 | 0 | 0 | 0.01 | 0 | 0 | 0 | 0 | 0.01 |
| SZS | 0 | 0 | 0 | 0 | 0.02 | 0 | 0 | 0 | 0 | 0.16 |
| XHY | 0 | 0 | 0 | 0 | 0.96 | 0 | 0 | 0 | 0 | 0.02 |





| | | | | | | | | | | |
|---|---|---|---|---|---|---|---|---|---|---|
| FG | 0 | 0 | 0 | 0 | 0.96 | 0 | 0 | 0 | 0 | 0.33 |
| TG | 0 | 0 | 0 | 0 | 0.33 | 0 | 0 | 0 | 0 | 0.21 |
| SMX | 0 | 0 | 0 | 0 | 0.55 | 0 | 0 | 0 | 0 | 0.36 |
| AS | 0 | 0 | 0 | 0 | 0.07 | 0 | 0.01 | 0 | 0 | 0.46 |
| LJ | 0 | 0 | 0 | 0 | 0 | 0 | 0 | 0 | 0 | 0.34 |
| ZT | 0 | 0 | 0 | 0 | 0.64 | 0 | 0 | 0 | 0 | 0.1 |
| ZJS | 0.07 | 0.08 | 0.08 | 0.08 | 0.92 | 0.1 | 0.09 | 0.06 | 0.08 | 0.89 |
| LZ | 0 | 0 | 0 | 0 | 0 | 0 | 0 | 0 | 0 | 0.32 |
| TDG | 0 | 0 | 0 | 0 | 0 | 0 | 0 | 0 | 0 | 0.27 |
| LM | 0 | 0 | 0 | 0 | 0 | 0 | 0 | 0 | 0 | 0.09 |
| LT | 0.01 | 0 | 0.01 | 0 | 0.93 | 0 | 0.02 | 0 | 0.01 | 0.79 |
| XY | 0.01 | 0.02 | 0 | 0.02 | 0.61 | 0 | 0.01 | 0 | 0.02 | 0.17 |

Further comparison was conducted on the preferred models of the two types at each station, with the null hypothesis (DM hypothesis) set as DAR-type model being superior to AR-GARCH-type. The optimal models for each station were ultimately

determined, with TNH, SZS, LJ, LZ, TDG, and LM stations favouring the FDTDAR-t model, the DAR-n model performing well at the ZJS station, and the remaining stations adopting the FDTDAR-n model.

## 5 Discussions

### 5.1 Properties of daily streamflow time series

The daily streamflow time series is influenced by natural and human factors, presenting seasonality, long-term persistence,

non-stationarity, and time-varying volatility (ARCH effect), which play a crucial role in simulation and prediction. However, the long-term persistence has been overlooked in most studies. The integer difference method is often used to address non-stationary issues to smooth the series (Myronidis et al., 2018; Yan et al., 2022), but it also eliminates the long-term memory information of daily streamflow, resulting in incomplete input information for the model and affecting the simulation and prediction accuracy. Some scholars (Mehdizadeh et al., 2019; O'Connell et al., 2016; Yang and Bowling, 2014) have also

studied long-term persistence, demonstrating its importance in predicting the daily streamflow process. However, those researchers have not conducted the second-order moment components, making it impossible to prove whether the ARCH effect is related to long-term persistence. In recent years, an increasing number of researchers (Dimitriadis et al., 2021; Graves et al., 2017; Grimaldi, 2004) have reported that the detected ARCH effect in the classic AR model may be suspicious due to the lack of consideration of long-term memory, and the results of the further constructed time-varying fluctuation model are thought-

provoking. This implies that the results obtained in existing studies on daily streamflow prediction by time series models may be biased due to an incomplete understanding of the characteristics of the research targets. Given the existing problems in the research, in addition to the seasonality and non-stationarity of the daily streamflow time series, this study also considers long-term persistence and time-varying volatility, which improved the prediction accuracy while ensuring the correctness of the ARCH effect.





In addition, time irreversibility is also an important property of streamflow time series, although it is negligible in other hydrological cycle components (e.g., precipitation, evapotranspiration) (Vavoulogiannis et al., 2021). Stationary processes in statistics are typically reversible over time, but the nonlinearity of underlying dynamic procedures is reflected in time irreversibility. River dynamics argues that the time irreversibility of runoff sequences mainly depends on different triggering factors in the formation process and exhibits nonlinear changes. Specifically, the increase in streamflow is largely determined

by short-term meteorological drivers, which are inherently the dynamic characteristics of precipitation, while groundwater dynamics respond to the streamflow reduction process (Serinaldi and Kilsby, 2016). Table 2 provides clear evidence of the nonlinear nature of daily streamflow, supporting the existence of time irreversibility. Threshold regression models such as TAR-GARCH and DTDAR have been successful in catching the nonlinear characteristics of daily streamflow time series, and numerous studies have confirmed their effectiveness in solving nonlinear problems. Therefore, the modeling approach used in

this study, incorporating long memory, thresholds, and time-varying fluctuations, is more suitable for describing daily streamflow time series..

**5.2 DAR-type model vs. AR-GARCH-type model**

From the general situation of the model, both the DAR and the AR-GARCH models can grasp the first- and second-order moment information of the daily streamflow time series, but the DAR model has a simpler structure. One of the most significant

differences between these models is that the second-order moment description of the GARCH model uses past conditional fluctuations, as well as past daily streamflow information, to estimate the current conditional variance. In contrast, the DAR model relies solely on past streamflow information. Previous period streamflow values and conditional fluctuations reflect the fluctuation aggregation and long-term behavior characteristics of the time series respectively. Nevertheless, most of the literature (Guo et al., 2021b; Modarres and Ouarda, 2013b; Pandey et al., 2019) indicated that the classic GARCH model is

insufficient in describing fluctuation aggregation effect, which significantly exists in hydrological time series (Fig. 2). Moreover, limited data availability in practice means that short-term volatility clustering is often prioritized over long-term volatility. To avoid long-term persistent interference, the daily streamflow time series is processed with an effective fractional difference approach before being used as the model input. Taken together, these constraints result in better prediction performance and accuracy for the DAR-type models than for the AR-GARCH-type models.

**5.3 Threshold regression model**

Geosciences acknowledge that data are the only reliable source of information for model construction, derivation, and prediction. The concepts of stationarity and non-stationarity are considered modeling options rather than properties of data because stochastic models are inherently mathematical constructs (Dimitriadis and Koutsoyiannis, 2018; Koutsoyiannis and Montanari, 2015). However, model properties such as randomness, determinism, stationarity, and non-stationarity should

always align with the data. Furthermore, stationarity is ergodic in practice, i.e., the selected sample data is representative of the entire stochastic process (Koutsoyiannis and Montanari, 2015). Meanwhile, the short-term behavior of hydrological





predictions can allow model structure to be inferred from sample data. These provide evidence for the rationality of using data to build models and predict in this study. In regression analysis, the stability of coefficient estimates is usually studied, but the external force disturbance that causes structural breaks in the streamflow time series cannot be avoided, which hinders the applicability of ordinary linear regression models.

Affected by multiple factors such as global climate change and intense human activities, the nonlinearity of the basin hydrological system has been enhanced. As early as the 1990s, Xia et al. (1997) had conducted in-depth and systematic research on nonlinear hydrological systems, and explored the nonlinear process of streamflow generation and transformation, as well as the nonlinear mechanism of streamflow formation and transformation from response units to watershed scales. He further proposed a new nonlinear model for streamflow simulation and prediction. In practice, the hydrological gain of streamflow magnitude generated by precipitation is closely related to highly nonlinear controlling factors (such as precipitation, underlying surface conditions, soil moisture, etc.), resulting in the system gain of streamflow generated by precipitation being non-steady, which is a nonlinear relationship. Therefore, linear mechanism process analysis is difficult to meet the needs of basin runoff simulation and prediction.

In view of the nonlinear changes in water resource systems (Wu et al. 2023), a series of multivariate analysis methods based on nonlinear relationships have been developed, where numerous conditioning factors are input into models to simulate streamflow (Song et al. 2022). These methods improve the simulation and prediction accuracy by comprehensively considering the influence of climate factors, soil properties, terrain conditions and vegetation types on streamflow (Tiwari et al. 2022). Regarding the land surface water balance, precipitation reaches the ground after being intercepted by vegetation leaves, and infiltrates through the soil or becomes surface streamflow. The water intercepted by the leaves and a portion of the water in the soil returns to the atmosphere through evaporation, and the soil moisture absorbed by vegetation for growth is transpired through the leaves, ultimately forming the cycle of atmospheric-land surface water. At present, coupled process and data-driven multivariate models are developing rapidly, which can take into account the mechanisms of streamflow and further enhance the simulation and prediction accuracy of single-type models (Zhong et al. 2023).

The univariate method constructed in this study is based on various performance characteristics of streamflow time series, reducing the uncertainty of input factors by generalizing the rainfall-runoff process in the basin, making practical applications more convenient. The threshold concept is used to explain the nonlinear variations in the series, which can more accurately describe the linear variation process of daily streamflow time series within different threshold intervals, and has strong robustness and applicability. Moreover, the threshold model is effective in capturing the asymmetric effects of rising and falling changes in daily streamflow time series, offering higher flexibility in grasping variation information compared to linear structure models, with high potential for simulation and prediction.

## 6 Conclusion

The nonlinear changes of the daily streamflow time series driven by the external environment cannot be ignored, and the strong non-stationarity and volatility, as well as its long-term memory, together lead to an increase in the difficulty of streamflow



prediction, which brings great challenges to the traditional time series models. To address this issue, this study introduced a novel DAR model and further proposed the DTDAR model, which considered these critical features and added the threshold, for improving the accuracy of daily streamflow time series prediction. The main conclusions of this study can be summarized as follows:

(1) The DAR model is a better alternative to the AR-GARCH model for predicting daily streamflow time series. Both the
linear structure DAR and the threshold DTDAR models outperform the AR-GARCH and TAR-GARCH models in terms of predictive ability.

(2) The nonlinear changes of the daily streamflow time series are reflected in multiple linear structures by adding the threshold, improving the accuracy of the single linear structure method. The NSE values of the DTDAR and TAR-GARCH models are higher than those of the DAR and AR-GARCH models by 0.056-0.229 and 0.032-0.431, respectively.

(3) The hypothesis type based on the t-distribution for residuals stands out in the prediction of daily streamflow time series for both FDTDAR and FTAR-GARCH models and is a better choice than the traditional normal distribution.

*Code and data availability.* The seasonally normalized and fractionally differencing daily streamflow data in this study can be downloaded from https://doi.org/10.6084/m9.figshare.26795140.v1. The codes for calculating results and plotting figures is
available upon request.

*Author contributions.* HW: conceptualization, methodology, software, formal analysis, data curation, writing – original draft, writing – review and editing, and visualization; SS: conceptualization, methodology, investigation, data curation, writing – review and editing, supervision, project administration, and funding acquisition; ZP: Formal analysis and writing – review and editing; GZ: conceptualization, methodology, investigation, data curation, writing – review and editing, supervision, project
administration, and funding acquisition.

*Competing interests.* The author has declared that there are no competing interests.

ther geographical representation in this paper. While Copernicus Publications makes every effort to include appropriate place names, the final responsibility lies with the authors.

*Financial support.* This study was supported financially by the National Natural Science Foundation of China (Grant No. 52079110), the Natural Science Foundation of Jiangsu Province (Grant No. BK20220590), and the China Postdoctoral Science Foundation under Grant Number 2024M752711, and the Priority Academic Program Development of Jiangsu Higher Education Institutions of China (PAPD).

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
