# Peer review of "DAR-type model based on "long memory-threshold" structure: a competitor for daily streamflow prediction under changing environment"

_EGUsphere, 2025_

## Author Comment (AC1)

I have read the paper, "DAR-type model based on "long memory-threshold" structure: a competitor for daily streamflow prediction under changing environment". Overall, the paper aims to develop and test a stochastic model for simulating daily streamflow, taking care of the nonlinearity, nonstationarity, and most importantly, the long-term memory of the streamflow. This is one of the few papers in the field of stochastic hydrology that has devoted greater attention to reproducing the long-term memory component of streamflow, which is really appreciated.

My major observation is that the paper is not sufficiently motivated, and the flow of the arguments in the paper is not smooth. For example, there are many times in the paper when an arbitrary number of statistical tests are being performed without any prior reasoning. The structure of section 2.3 does not clearly give enough reason why the current modeling paradigm is failing to reproduce the nonlinear, non-stationary models that fail to reproduce the long-term memory properties of the streamflow. Further, this section does not provide enough evidence to go with the FDTDAR model. There are many figures in the paper which is more suitable in the supplementary file rather than the main manuscript.

**Responses:** Thank you very much for your constructive and insightful comments. We truly appreciate your careful reading and recognition of the novelty of our work, especially your appreciation for our focus on long-term memory modeling in streamflow simulation. We have carefully considered your main observation that the motivation and logical flow of the paper need improvement, and that Section 2.3 lacks clarity and sufficient justification, including the relocation of figures to the supplementary files. In response, we have followed the comments and suggestions to refine this work. We highlight revisions with bold and blue text in the latest submitted manuscript and list them following.

2.3 Daily streamflow time series characteristics
Before initiating the modeling procedure, it is essential to understand the inherent statistical properties of daily streamflow time series. The primary objective of this section is to conduct a comprehensive diagnostic analysis to evaluate four key characteristics of time series: long-term persistence, non-stationarity, ARCH effects, and nonlinearity (Figure 2). These features are critical in determining the suitability

and effectiveness of various modeling approaches. Moreover, understanding their interconnections provides important theoretical support for the logical consistency and robustness of the subsequent modeling framework (Figure 2).

The daily streamflow time series exhibits seasonal characteristics due to the cyclical influence of the four seasons. Therefore, deseasonalization is a necessary preprocessing step before modeling, with the specific method detailed in Section 3.1. In this section, the presence of long-term memory in the daily streamflow series is identified based on the Hurst exponent (H) calculated using the rescaled range (R/S) analysis method and the trend of the autocorrelation function (ACF), and the influence of seasonality on long-term memory is also assessed (Lo, 1991). On this basis, the stationarity of the series is evaluated using three unit root tests: the Augmented Dickey-Fuller (ADF), Phillips-Perron (PP), and Kwiatkowski-Phillips-Schmidt-Shin (KPSS) tests (Dickey and Fuller, 1981; Kwiatkowski et al., 1992; Peter and Perron, 1988). The null hypotheses of ADF and PP tests assume that the time series has a unit root, while the KPSS test considers the null hypothesis as the time series being a stationary process. Due to the tendency of unit root tests to favor the null hypothesis, to rigorously determine the non-stationarity of the daily streamflow time series, if any of the three methods indicates the presence of a unit root process, then the daily streamflow time series at that station is considered non-stationary. Stationarity is a fundamental prerequisite for model construction. The method of achieving stationarity differs depending on the presence of long memory: non-long-memory series are differenced using integer orders, while long-memory series require fractional differencing. Subsequently, the Ljung-Box (LB) and Lagrange Multiplier (LM) tests are employed to detect the presence of ARCH effects, which is the most critical link in choosing to build a heteroscedastic model in this study (Ljung and Box, 1978). The ARCH effect reflects the limitations of linear regression models and highlights the necessity of characterizing the variation in second-order moments. Some studies have suggested a close relationship between ARCH effects and long-term memory in hydrological time series, neglecting long-term memory may lead to spurious detection of ARCH effects (Wang et al., 2023b; Wang et al., 2023c). Therefore, this section compares the LB test results of two linear models: a standard integer-order autoregressive model and a long-memory autoregressive (FAR) model based on fractional differencing. Finally, the BDS test is applied to examine whether the series exhibits nonlinear dynamics (Broock et al., 1996). Since most existing heteroscedastic models are constructed based on linear structures, if the series demonstrates significant nonlinearity, it may be necessary to improve the model expression accordingly.

[Figure]

Figure 2: Daily streamflow time series characteristics test and application method.
The results of various diagnostic tests on the daily streamflow time series are presented in Figure 4, Figure S2, and Tables S1–S3. The autocorrelation function (Figure 3) and the Hurst exponent results (Table S1) indicate that the daily streamflow series exhibits long-ter memory characteristics, and that seasonality weakens the strength of this memory. The results of the 3 unit root tests (Table S2) reveal that the series is non-stationary. Considering its long-memory properties, fractional differencing is adopted to achieve stationarity, which satisfies the model assumptions. Figure S2 shows the Ljung-Box (LB) test results for the residuals of two linear models. The residuals of both models exhibit no significant autocorrelation, indicating that the model structures are appropriate. However, the squared residuals display significant autocorrelation, reflecting the presence of ARCH effects. Moreover, the ARCH effect becomes more pronounced when long memory is taken into account. The LM test yields consistent results. The nonlinearity test results (Table S3) suggest that the daily streamflow time series exhibits significant nonlinear behavior, implying that the model structure in this study should be modified to accommodate nonlinear dynamics.

[Figure]

Figure 3: Autocorrelation coefficient (ACF) of daily streamflow time series.
The diagnostic results presented above provide critical insights into the statistical characteristics of the daily streamflow time series, including long-term memory, non-stationarity, heteroscedasticity, and nonlinearity. These findings form a solid foundation for selecting appropriate modeling strategies in the subsequent sections. Specifically, the detection of long-term memory and non-stationarity justifies the use of fractional differencing methods to ensure model validity. The presence of ARCH effects highlights the need for models capable of capturing time-varying volatility, while the identified nonlinear dynamics suggest that linear structural models may be insufficient. DAR-type models have emerged as strong competitors to the widely used heteroscedastic GARCH models, due to their simple mathematical formulation, ease of parameter estimation, and ability to accurately characterize real-world data. Based on the results of this section, the next section introduces an enhanced modeling framework (FDTDAR model) based on the DAR family, which integrates long-memory components, heteroscedasticity, and nonlinear dynamics to better capture the complex behavior of daily streamflow.

The following comments need to be addressed to improve the structure of the paper and the overall motivation behind this work.

**Major comment:**

- Line 120-125: How is the Hurst exponent estimated? Based on the information provided in Table 1, the length of the time series is short enough to estimate a stable value of H. Additionally, there is no uncertainty measure of the H estimates provided. This is a very serious concern. If H is not statistically significant, then it is not a long-term persistence process. In order to confirm the existence of long-term persistence, the nature of decay of the autocorrelation function must follow a type of power-law, as long-term persistence is a scale-free entity.

**Responses:** Thanks for your comment. We agree that accurate estimation of the Hurst exponent (H) is critical to validate the presence of long-term persistence, and we appreciate the opportunity to clarify and improve this aspect of the paper.

(1) Method for estimating the Hurst exponent:

We have clarified the calculation method of the H index in the description on page 6, lines 122-124 of the manuscript. Specifically, we used the rescaled range analysis (R/S) method, which is widely used and robust in detecting long-term correlations in hydrological time series. The specific description in the manuscript is " In this section, the presence of long-term memory in the daily streamflow series is identified based on the Hurst exponent (H) calculated using the rescaled range (R/S) analysis method and the trend of the autocorrelation function (ACF)". A supplementary file has also been added to the paper, in which we describe the calculation process of the R/S method. For details, please refer to the supplementary file page 2, lines 5-15: "In this study, the rescaled range (R/S) analysis method is employed to calculate the Hurst exponent (H) of daily streamflow time series from 15 hydrology station. For time series $X_t$ of length $T$ with $t = 0,1,\cdots,T$, the R/S analysis proceeds as follow:

First, calculate the increments of the time series $x_t = X_t - X_{t-1}$ with $t = 0,1,\cdots,T$ and divides them into $N$ adjacent sub-interval $I_n$, for $n = 1,\cdots,N$, of length $v$ while $Nv = T$. For each sub-interval the mean value is computed, and a cumulative deviation series $y_t$ is constructed by accumulating deviations from this mean.

Subsequently, the range $R_{I_n} = max_{I_n}(y_t) - min_{I_n}(y_t)$ of the cumulative deviation series $y_t$ and a standard deviation $S_{I_n}$ of the original increment time series for each

sub- interval are calculated. Each range is divided by its corresponding standard deviation to obtain the standardized rescaled range. Finally, the average rescaled range $(R/S)_v$ for the given sub-period length is calculated by averaging across all sub-intervals. Rescaled ranges scale as $(R/S)_v \propto v^H$.".

(2) Justification for statistical stability:

In the revised manuscript, we have clarified the method used to estimate the Hurst exponent. Specifically, we employed the rescaled range (R/S) method, which is widely used in hydrology and time series analysis due to its robustness in detecting long-term persistence, even in relatively short records. However, to illustrate the uncertainty measure of the H value and enhance the robustness of its estimation, we have included the confidence intervals and significance tests of the H values for the daily streamflow series from the 15 hydrological stations in the supplementary material. These were obtained through Monte Carlo simulations. Specifically, for each station, we generated 1,000 synthetic Gaussian white noise sequences (with H = 0.5) of the same length as the observed streamflow series. Under the null hypothesis of no long-term persistence, we calculated the 95% confidence intervals of the H estimates. If the observed H value falls outside this range (p < 0.05), it indicates statistically significant long-term persistence in the time series. The detailed results are presented in the supplementary material (page 3, lines 20-40), including the following description: "To assess the statistical significance of the H estimated using the R/S method, a confidence interval for the test statistic was constructed based on the Monte Carlo simulation approach. Under the null hypothesis, the time series is assumed to exhibit no long-term memory, i.e., H=0.5, corresponding to a Gaussian white noise process. Specifically, for each hydrological station, 1,000 synthetic Gaussian white noise sequences of the same length as the observed daily streamflow series were generated. The R/S analysis was then applied to each simulated series to estimate the Hurst exponent, and the 95% confidence interval was derived from the resulting distribution. If the H value of the observed series lies outside this interval (at a significance level of $p<0.05$), the time series is considered to exhibit statistically significant long-term memory. As shown in Table S1, all 15 hydrological stations demonstrate significant long-term persistence in their daily

streamflow time series.

**Table S1**. Significance test of H value of daily streamflow time series at 15 hydrological stations

| Station | 2.50% | 97.50% | H | p | Station | 2.50% | 97.50% | H | p |
|---------|-------|--------|------|-------|---------|-------|--------|------|-------|
| TNH | 0.49 | 0.59 | 0.82 | <0.05 | ZT | 0.49 | 0.59 | 0.69 | <0.05 |
| SZS | 0.49 | 0.59 | 0.88 | <0.05 | ZJS | 0.48 | 0.59 | 0.72 | <0.05 |
| XHY | 0.49 | 0.59 | 0.91 | <0.05 | LZ | 0.49 | 0.60 | 0.92 | <0.05 |
| FG | 0.48 | 0.59 | 0.88 | <0.05 | TDG | 0.49 | 0.59 | 0.87 | <0.05 |
| TG | 0.49 | 0.59 | 0.87 | <0.05 | LM | 0.49 | 0.59 | 0.89 | <0.05 |
| SMX | 0.48 | 0.59 | 0.87 | <0.05 | LT | 0.48 | 0.59 | 0.76 | <0.05 |
| AS | 0.48 | 0.59 | 0.84 | <0.05 | XY | 0.48 | 0.58 | 0.77 | <0.05 |
| LJ | 0.49 | 0.59 | 0.85 | <0.05 | | | | | |

".

(3) Assessment of long-term persistence via autocorrelation decay:

As suggested, we have included an analysis of the decay of the autocorrelation function (ACF) in our current manuscript (shown in Figure 3 on page 8). The results show that the ACF decays in a power-law manner, rather than exhibiting the exponential decay characteristic of short-term memory processes. This pattern provides additional evidence supporting the presence of long-term persistence in the daily streamflow series.

- Section 2.3: This section, in the current form, is the most confusing part in section 2. It portrays different tests on streamflow time series, identifies non-stationarity, confirms the general properties of a white noise process, and examines the contribution of long-term memory to improve the simulation capability of daily streamflow. There are many tests performed here, without enough motivation. It is recommended to reconstruct this section. First, try to state what the overall aim or motivation is for the modeling exercise. Second, with the help of a flowchart, show what hypothesis needs to be tested before going to the modeling exercise. In order to do such hypothesis testing, state the relevant tests with appropriate references. Finally, crisply conclude what is learnt through this modeling exercise and state the

next steps to achieve the objective of the paper.

**Responses:** We sincerely thank the reviewer for the constructive comments on Section 2.3. We fully agree that the original version lacked clarity in structure and motivation, and could potentially confuse the reader due to the number of statistical tests presented without a clearly stated objective. In response, we have substantially revised Section 2.3 to improve its coherence and readability. The revisions include the following key improvements:

(1) Clarification of Motivation and Objective:

At the beginning of Section 2.3, we have added a clear statement of purpose, explaining that the main objective is to evaluate the intrinsic statistical properties of the daily streamflow series-including stationarity, randomness, and long-term persistence-before proceeding with any modeling. This diagnostic step is essential to inform the selection of appropriate modeling approaches and ensure that underlying assumptions are satisfied. This is specifically reflected in the revised manuscript on page 6, lines 115-120, which are described as: " Before initiating the modeling procedure, it is essential to understand the inherent statistical properties of daily streamflow time series. The primary objective of this section is to conduct a comprehensive diagnostic analysis to evaluate four key characteristics of time series: long-term persistence, non-stationarity, ARCH effects, and nonlinearity (Figure 2). These features are critical in determining the suitability and effectiveness of various modeling approaches. Moreover, understanding their interconnections provides important theoretical support for the logical consistency and robustness of the subsequent modeling framework (Figure 2).".

(2) Introduction of a Hypothesis Testing Framework:

We have added a structured hypothesis testing framework to the revised manuscript, supported by a flowchart that outlines the logical flow of the analysis. Relevant references have also been added for each test. The adjustments to this section are detailed in pages 6-7, lines 121-144 of the revised manuscript, which are described as follows: " The daily streamflow time series exhibits seasonal characteristics due to the cyclical influence of the four seasons. Therefore, deseasonalization is a necessary

preprocessing step before modeling, with the specific method detailed in Section 3.1. In this section, the presence of long-term memory in the daily streamflow series is identified based on the Hurst exponent (H) calculated using the rescaled range (R/S) analysis method and the trend of the autocorrelation function (ACF), and the influence of seasonality on long-term memory is also assessed (Lo, 1991). On this basis, the stationarity of the series is evaluated using three unit root tests: the Augmented Dickey-Fuller (ADF), Phillips-Perron (PP), and Kwiatkowski-Phillips-Schmidt-Shin (KPSS) tests (Dickey and Fuller, 1981; Kwiatkowski et al., 1992; Peter and Perron, 1988). The null hypotheses of ADF and PP tests assume that the time series has a unit root, while the KPSS test considers the null hypothesis as the time series being a stationary process. Due to the tendency of unit root tests to favor the null hypothesis, to rigorously determine the non-stationarity of the daily streamflow time series, if any of the three methods indicates the presence of a unit root process, then the daily streamflow time series at that station is considered non-stationary. Stationarity is a fundamental prerequisite for model construction. The method of achieving stationarity differs depending on the presence of long memory: non-long-memory series are differenced using integer orders, while long-memory series require fractional differencing. Subsequently, the Ljung-Box (LB) and Lagrange Multiplier (LM) tests are employed to detect the presence of ARCH effects, which is the most critical link in choosing to build a heteroscedastic model in this study (Ljung and Box, 1978). The ARCH effect reflects the limitations of linear regression models and highlights the necessity of characterizing the variation in second-order moments. Some studies have suggested a close relationship between ARCH effects and long-term memory in hydrological time series, neglecting long-term memory may lead to spurious detection of ARCH effects (). Therefore, this section compares the LB test results of two linear models: a standard integer-order autoregressive model and a long-memory autoregressive (FAR) model based on fractional differencing. Finally, the BDS test is applied to examine whether the series exhibits nonlinear dynamics (Broock et al., 1996). Since most existing heteroscedastic models are constructed based on linear structures, if the series demonstrates significant nonlinearity, it may be necessary to improve the model

expression accordingly.

[Figure]

**Figure 2: Daily streamflow time series characteristics test and application method.**".

(3) Refinement of Text and Interpretation:

We have rewritten the section to describe each test concisely, with a brief explanation of its purpose and relevance. The results are interpreted in the context of the modeling objective, ensuring that each test logically contributes to the decision-making process.

(4) Summary and Transition:

At the end of Section 2.3, a concise summary paragraph has been added to synthesize the key findings-namely, that the streamflow series is non-stationary, non-random, and exhibits significant long-term memory. This conclusion justifies the subsequent use of long-memory modeling approaches. For details, see page 8, lines 145-155 of the revised manuscript, specifically describing "The results of various diagnostic tests on the daily streamflow time series are presented in Figure 4, Figure S2, and Tables S1–S3. The autocorrelation function (Figure 3) and the Hurst exponent results (Table S1) indicate that the daily streamflow series exhibits long-ter memory characteristics, and that seasonality weakens the strength of this memory. The results of the 3 unit root tests (Table S2) reveal that the series is non-stationary. Considering its long-memory properties, fractional differencing is adopted to achieve stationarity, which satisfies the model assumptions. Figure S2 shows the Ljung-Box (LB) test results for the residuals

of two linear models. The residuals of both models exhibit no significant autocorrelation, indicating that the model structures are appropriate. However, the squared residuals display significant autocorrelation, reflecting the presence of ARCH effects. Moreover, the ARCH effect becomes more pronounced when long memory is taken into account. The LM test yields consistent results. The nonlinearity test results (Table S3) suggest that the daily streamflow time series exhibits significant nonlinear behavior, implying that the model structure in this study should be modified to accommodate nonlinear dynamics.". The transition to the next section clearly states how these findings inform the modeling framework adopted in the remainder of the study. In the revised manuscript, lines 158-167 on pages 8-9 are described as: "The diagnostic results presented above provide critical insights into the statistical characteristics of the daily streamflow time series, including long-term memory, non-stationarity, heteroscedasticity, and nonlinearity. These findings form a solid foundation for selecting appropriate modeling strategies in the subsequent sections. Specifically, the detection of long-term memory and non-stationarity justifies the use of fractional differencing methods to ensure model validity. The presence of ARCH effects highlights the need for models capable of capturing time-varying volatility, while the identified nonlinear dynamics suggest that linear structural models may be insufficient. DAR-type models have emerged as strong competitors to the widely used heteroscedastic GARCH models, due to their simple mathematical formulation, ease of parameter estimation, and ability to accurately characterize real-world data. Based on the results of this section, the next section introduces an enhanced modeling framework (FDTDAR model) based on the DAR family, which integrates long-memory components, heteroscedasticity, and nonlinear dynamics to better capture the complex behavior of daily streamflow.".

- Before going to section 4, it is recommended to give an illustration of the model selection, parameter estimation, and testing the residuals of the FDTDAR model for a simulation from a standard model. Give a flowchart for this entire model-building process and diagrams related to the key results.

**Responses:** We thank the reviewer for this valuable suggestion. In response, we have added a detailed illustration of the model-building process for the FDTDAR model prior to Section 4. This new content serves as a bridge between the diagnostic analysis and the model application phase, and aims to enhance the transparency of our modeling framework.

(1) Model Selection and Justification:

We added a brief summary at the beginning of Section 3.1 in the revised manuscript, explaining why the FDTDAR model is constructed based on the statistical characteristics of the determined daily runoff time series. The description of lines 179-188 on pages 9-10 in the revised manuscript is: "As revealed by the analysis in Section 2.3, the daily streamflow time series exhibits long-term memory, non-stationarity, conditional heteroscedasticity, and nonlinear characteristics. Therefore, these properties should be comprehensively considered in the model construction process. The model proposed in this study, referred to as the FDTDAR model (Fractionally differenced dual-threshold double autoregressive model), is an extension of the ordinary double autoregressive (DAR) model-a type of heteroscedasticity model. Specifically, the long memory characteristics of daily streamflow are described using the fractional difference method. On the premise of satisfying stationarity, the threshold idea is considered in the DAR model, that is, in the process of characterizing the first-order moment and the second-order moment, the threshold is introduced to segment, so as to express the nonlinear changes of the daily streamflow series. This processing method is inspired by the threshold autoregression (TAR) model. The FDTDAR model is therefore capable of accommodating the coexistence of multiple complex features in the daily streamflow series. The mathematical formulation of the model is presented as follows:".

(2) Parameter Estimation:

In the revised manuscript, we have clarified the parameter estimation method used for the proposed FDTDAR model at the beginning of Section 3.3 "Parameter Estimation of the FDTDAR Model." We have also explicitly stated that two types of residual distributions (normal and Student's t) are considered in the estimation process. This is

described on Page 11, Lines 226-227 of the revised manuscript, as follows: "The parameters of the FDTDAR model are estimated using the quasi-maximum likelihood estimation (QMLE) method, considering both the Gaussian distribution (FDTDAR-n) and the student's t distribution (FDTDAR-t) for the residuals.".

Moreover, although the original version of Section 3.3 already included the maximum likelihood estimation formulas for the FDTDAR model under both distributional assumptions, it lacked details regarding the software implementation and the settings of the initial values of the parameters in the estimation process. These aspects have now been added in the revised manuscript. Please refer to Page 13, Lines 280–288 for the newly added description: "In this study, the quasi-maximum likelihood estimation under both residual distributions was implemented using the *nlminb* function in R version 4.4.1, which is an efficient numerical optimization tool. It returns the negative value of the likelihood function; therefore, the actual likelihood value used in the estimation is the negative of the function's output. For both residual distribution scenarios, the model orders ($p_{11}$, $p_{21}$, $p_{12}$, $p_{22}$, $q_{11}$, $q_{12}$, $q_{21}$, and $q_{22}$; $p_1$, $p_2$, $q_1$ and $q_2$) were exhaustively searched in the range of 0 to 10. The initial values of the parameters were determined based on the autocorrelation functions at different lag orders up to the corresponding model order. Additionally, the *nlminb* function allows for setting constraints on parameter values, which were specified based on the theoretical requirements of the model. Specifically, the parameters associated with the first-order moments were allowed to vary over the entire real line, while the parameters related to the second-order moments were constrained to be greater than 0.".

(3) Residual Diagnostics:

We added an introduction to model residual diagnosis in the model evaluation method section (Section 3.6). For details, please see page 15, lines 329-332 of the revised manuscript, which reads: "The residual diagnostics of the model were conducted using the autocorrelation function (ACF), by examining whether the residual autocorrelations at various lag times fall within the 95% confidence interval around 0. If ACF values lie within this interval, the residuals can be considered approximately white noise, indicating that the model has effectively captured the variation structure of the time

series.".

(4) Model building flowchart:

An overview of the process of building the model in this study is shown in Figure 4.

[Figure]

Figure 4: Model building overview diagram

The key results related to the long-term memory-heteroscedastic-nonlinear structure models have been presented in Section 4.1. Specifically, the selection of model orders and threshold values are shown in Figure 5 and Table 2, respectively. Figures 6 and 7 present the comparison of predictive performance and the scatter distribution of forecasts for the FDTDAR and FTAR-GARCH models under two types of residual distributions. Figure 8 displays the ACF test results of the model residuals.

- Section 3.5: Why the FTAR-GARCH model? The previous sections were devoted to a finer understanding of the FDTDAR model. Suddenly, in this section, a new modeling framework is added without any prior motivation/reasons. Please clarify this point.

**Responses:** We appreciate the reviewer's insightful comment regarding the sudden introduction of the FTAR-GARCH model in Section 3.5. To address this concern, we have revised the manuscript to clarify the rationale for introducing the FTAR-GARCH

model. As mentioned in the Introduction, AR-GARCH type models are popular and widely recognized approaches for modeling heteroscedasticity in time series. However, the complex structure of these combined models often leads to difficulties in parameter estimation, and the strict parameter constraints of AR-type models further limit their applicability in real-world scenarios. In contrast, DAR-type models have recently emerged as a promising alternative for modeling heteroscedasticity. Due to their simpler mathematical formulation and more flexible parameter settings, they are easier to implement and generally provide more reliable results. The FTAR-GARCH model introduced in Section 3.5 retains the basic structure of the traditional AR-GARCH framework, but similar to the FDTDAR model, it adds elements that capture long-memory behavior (via fractional differencing) and nonlinear dynamics (through threshold effects). Therefore, our primary motivation for introducing the FTAR-GARCH model is to use it as a benchmark for evaluating the predictive performance of the proposed FDTDAR model.

To improve the coherence of the manuscript, we also added a brief introductory paragraph at the beginning of Section 3.5 to explain the motivation for constructing the FTAR-GARCH model. This clarification has been added to the beginning of Section 3.5 (see revised manuscript, Page14, Lines 307-308), which now reads: "To evaluate the performance of the proposed FDTDAR model, we compare it with the FTAR-GARCH model, a widely used long-memory heteroscedastic model that also incorporates threshold effects.".

**Other comments:**

- There are many figures and tables in the main manuscript that can be moved to the supplementary section, as they support the model development process. For example, table 2, figure 4, and figure 2, table 3.

**Responses:** Thanks for your suggestion. In the revised manuscript, we have moved Table 2, Figure 4 and Figure 2, Table 3 in the original version to Table S2, Figure S2 and Figure S1, Table S3 in the supplementary file.

- Section 2.3: Daily streamflow time series characteristics and their linkage relationships – the title is not conveying any specific element/property. What is meant by linkage relationships? How is this link estimated? What are the variables considered for the link?

**Responses:** Thank you for pointing out the ambiguity in the original title and content of Section 2.3. We acknowledge that the phrase "Daily streamflow time series characteristics and their linkage relationships" was too vague and failed to clearly convey the focus of the section. In the original version, the intention was to examine whether the daily streamflow time series exhibits long-term memory, non-stationarity, ARCH effects, and nonlinearity, as well as whether these properties interact with each other (i.e., linkage relationships). The section did not employ conventional correlation measures involving variable pairs. For example, the influence of seasonality on long-term memory was assessed by comparing the Hurst exponent values and autocorrelation function (ACF) behavior of the original series and the deseasonalized series. The observation that the deseasonalized series exhibits higher H values and a slower ACF decay suggests that removing seasonality enhances the strength of long-term memory. Similarly, the effect of long-term memory on the presence of ARCH effects (conditional heteroskedasticity) was also evaluated. We agree that the use of the term "linkage relationships" was imprecise. To avoid confusion, we have removed this term from the title and revised the title of Section 2.3 to "Daily streamflow time series characteristics".

- Line 130-135: There is no motivation for doing all sorts of statistical tests. In the previous section, the discussion was focused on autocorrelation, but suddenly it shifted to nonstationarity without giving much motivation for why such an analysis is needed.

**Responses:** We agree that the transition between different statistical tests in the original version was abrupt and lacked a clear motivating context. We have substantially revised and reorganized Section 2.3 in the updated manuscript. The revised section now consists of four structured paragraphs: (1) an overall introduction and objective of the section, (2) an introduction of the methods used to examine each statistical property, (3)

a summary of the diagnostic results, and (4) a transitional paragraph leading into the subsequent modeling section. In the first paragraph, we clearly state that the objective of this diagnostic analysis is to determine the statistical characteristics of the daily streamflow series, such as long-term memory, nonstationarity, conditional heteroskedasticity, and nonlinearity-which are critical for selecting an appropriate modeling framework. Therefore, after identifying long memory based on the autocorrelation function, we proceed to diagnose nonstationarity and other statistical features. The abrupt transitions present in the original version have been addressed and resolved through the new structured and logically coherent revision.

- Line 120-125: How the deseasonalization is performed here. Please provide the mathematical details of the process.

**Responses:** We thank the reviewer for this helpful suggestion. In the revised manuscript, we have restructured Section 2.3 according to the reviewer's other suggestions. As part of this revision, we have clearly indicated where the mathematical details of the deseasonalization process are provided. Specifically, these details are included in Section 3.1. Please refer to Page 6, Lines 121-122 of the revised manuscript, which state: "The daily streamflow time series exhibits seasonal characteristics due to the cyclical influence of the four seasons. Therefore, deseasonalization is a necessary preprocessing step before modeling, with the specific method detailed in Section 3.1.".

- Figure 2: The discharge is shown in $m^3/s$ with a different y-axis. Please show the plots in mm/day, so that the flow magnitudes and other patterns can be compared visually across all the catchments.

**Responses:** We have moved Figure 2 from the original manuscript to the supplementary material as Figure S1. However, we fully acknowledge the reviewer's valid concern regarding the inconsistency in the y-axis scales, which indeed makes it difficult to visually compare daily streamflow magnitudes across all hydrological stations. To address this issue while maintaining consistency with the units used in the subsequent model comparison figures, we retained the current unit ($m^3/s$) but adjusted the y-axis

display range to be consistent across all subplots. This revision resolves the concern raised and ensures logical consistency throughout the manuscript.

- Line 96: The Length of the basin is 5464 km, is it the length of the main channel/river? Please provide the catchment area.

**Responses:** Thanks for your comments. We have clarified in the revised manuscript that the length of the Yellow River mainstream is 5464 kilometers, and added the catchment area of the basin. Please see page 4, lines 97-98 of the revised manuscript for details, which is specifically described as "The research area of this study (Figure 1) is the Yellow River Basin in China, which has a main stream length of approximately 5464 km and a catchment area of about 795,000 km², ranking second in China and fifth in the world. ".

- Section 3.1: "mu_m and sigma_m are seasonal mean and variance" – I think this is not correct. It is mentioned in equation 1 that $m$ denotes day of the year where $n$ denotes the year. Therefore, the variable x_nm denotes the value of streamflow on the mth day of the nth year. So if the average is taken across all the years (as it is mentioned n=1,2,..,N), mu_m should be the average annual streamflow, not the seasonal streamflow as it is now mentioned. Please clarify this. The same is with the variance, sigma_m.

**Responses:** Thanks for your comments. We have changed the expression to "$\mu^m$ and $\sigma^m$ are the mean and variance of the variable for each calendar day, respectively." in the revised manuscript on page 9, lines 175-176.

- Section 2.3: Strong motivation for why the DAR type model is needed in streamflow simulation can be discussed here with some numerical cases.

**Responses:** Thank you for this helpful suggestion. We agree that providing a stronger motivation for using DAR-type models in streamflow simulation would enhance the clarity and impact of Section 2.3. In the revised manuscript, we have expanded the discussion to more clearly explain the rationale behind adopting the DAR framework. Specifically, we emphasize that through various tests, daily streamflow series often

exhibit long-term memory behavior, conditional heteroscedasticity, and nonlinear dynamics, which are not well captured by traditional linear models. DAR-type models provide a concise and easy-to-interpret heteroscedasticity structure, which has the potential to surpass GARCH-type models in real work. We describe this in the transition paragraph in Section 2.3, which also leads to the improved framework of the DAR model in the next section (FDTDAR model), which includes the characterization of long-term memory and nonlinear changes. The specific description is: "The diagnostic results presented above provide critical insights into the statistical characteristics of the daily streamflow time series, including long-term memory, non-stationarity, heteroscedasticity, and nonlinearity. These findings form a solid foundation for selecting appropriate modeling strategies in the subsequent sections. Specifically, the detection of long-term memory and non-stationarity justifies the use of fractional differencing methods to ensure model validity. The presence of ARCH effects highlights the need for models capable of capturing time-varying volatility, while the identified nonlinear dynamics suggest that linear structural models may be insufficient. DAR-type models have emerged as strong competitors to the widely used heteroscedastic GARCH models, due to their simple mathematical formulation, ease of parameter estimation, and ability to accurately characterize real-world data. Based on the results of this section, the next section introduces an enhanced modeling framework (FDTDAR model) based on the DAR family, which integrates long-memory components, heteroscedasticity, and nonlinear dynamics to better capture the complex behavior of daily streamflow." on pages 8-9, lines 158-167 of the revised manuscript.

- Section 4.1 is not there.

**Responses:** Thanks for your discovery of the problem in the manuscript. In the revised manuscript, sections "4.2" and "4.3" have been re-adjusted to "4.1" and "4.2", respectively.

---

## Author Comment (AC2)

Summary:

This manuscript proposes an innovative modelling approach - the dual-threshold double autoregressive (DTDAR) model - designed to improve the prediction of daily streamflow under non-stationarity, long memory, and non-linearity in the field of hydrology. By integrating fractional differencing with a threshold-based structure in both the first- and second-order moments, the authors develop a long memory-threshold framework (FDTDAR) that is shown to outperform conventional models (AR-GARCH, TAR-GARCH) at multiple stations across the Yellow River Basin.

**General remarks:**

1. The manuscript is well structured, comprehensive in analysis, and the methodology is methodically laid out. The proposed approach makes a meaningful contribution to the field of stochastic hydrological modelling and represents a promising alternative to existing linear and GARCH-based models. However, several aspects require clarification or revision before the manuscript can be recommended for publication.

**Responses:** We sincerely appreciate the reviewer's positive and encouraging evaluation of our work. We are grateful for your recognition of the manuscript's structure, methodological clarity, and contribution to the field of stochastic hydrological modeling. Your thoughtful comments have helped us to further improve the quality and clarity of the manuscript. We have carefully revised the manuscript in response to your questions, which are highlighted in yellow in the latest revision submitted.

2. While the DTDAR/FDTDAR model is a novel approach, it is structurally complex, with numerous parameters and thresholds. The paper would benefit from a clearer explanation of how parameter identifiability, estimation convergence, and computational burden are handled. Practitioners will benefit from a discussion on model tractability and software implementation.

**Responses:** We thank the reviewer for this important and constructive comment. We agree that the DTDAR/FDTDAR model, while novel in its structure, involves a relatively large number of parameters and thresholds, which raises concerns regarding

parameter identifiability, convergence, and computational cost. To address this, we have revised the manuscript to provide a more detailed explanation of the estimation procedure.

We have clarified in the revised manuscript the constraints and initialization strategies adopted during the estimation process to ensure parameter identifiability. In addition, we have added descriptions of the convergence criteria and diagnostic methods used in the numerical optimization, as well as details on the software implementation of the model. These revisions are presented on ==page 13, lines 280-288 of== the revised manuscript, specifically as follows: "In this study, the quasi-maximum likelihood estimation under both residual distributions was implemented using the *nlminb* function in R version 4.4.1, which is an efficient numerical optimization tool. It returns the negative value of the likelihood function; therefore, the actual likelihood value used in the estimation is the negative of the function's output. For both residual distribution scenarios, the model orders ($p_{11}$, $p_{21}$, $p_{12}$, $p_{22}$, $q_{11}$, $q_{12}$, $q_{21}$, and $q_{22}$; $p_1$, $p_2$, $q_1$ and $q_2$) were exhaustively searched in the range of 0 to 10. The initial values of the parameters were determined based on the autocorrelation functions at different lag orders up to the corresponding model order. Additionally, the *nlminb* function allows for setting constraints on parameter values, which were specified based on the theoretical requirements of the model. Specifically, the parameters associated with the first-order moments were allowed to vary over the entire real line, while the parameters related to the second-order moments were constrained to be greater than 0.".

3. The manuscript focuses primarily on comparisons with AR-GARCH and TAR-GARCH models. While these are relevant, the absence of modern nonlinear or machine learning models (e.g., LSTM, hybrid deep learning models) in the comparison set limits the extent to which the results can be considered broadly applicable. Even if not implemented, a discussion acknowledging this limitation and the rationale for focusing on DAR-type models would be appropriate.

**Responses:** We thank the reviewer for this insightful comment. We fully acknowledge that the comparison in our study has been primarily limited to traditional linear and

GARCH-based models, such as AR-GARCH and TAR-GARCH. While these models are relevant and widely used benchmarks in stochastic hydrological modeling, we agree that the absence of modern nonlinear or machine learning approaches, such as LSTM or hybrid deep learning models, may limit the generalizability of our conclusions. Our primary focus in this study is to explore the potential of DAR-type models, particularly the proposed FDTDAR structure, as a parsimonious yet effective alternative to time series models. The rationale for this focus lies in the interpretability, mathematical transparency, and relatively low data and computational requirements of DAR-type models, which are particularly important in hydrological applications with limited data availability or operational constraints. We have added a comparison paragraph with modern learning-like models in the "5. Discussion" section of the revised manuscript, although such models are not presented in this paper. Please refer to page 27, lines 516-544 of the revised manuscript for details, which reads:

"5.4 Limitations of FDTDAR models

Compared with modern machine learning models (such as long short-term memory (LSTM) networks or hybrid deep learning architectures), the DAR-type models offer a simpler and more interpretable approach to time series modeling. They are particularly advantageous in scenarios where domain interpretability, limited sample sizes, and computational efficiency are critical. Their transparent structure facilitates parameter estimation, theoretical analysis, and diagnostic evaluation, making them especially suitable for hydrological applications that typically involve noisy data, sparse observations, and operational constraints, thereby enhancing their robustness and reliability in real-world settings (Li et al., 2019; Ling, 2007).

However, the proposed FDTDAR model also has inherent limitations. First, although it can effectively capture regime-switching behavior and certain nonlinear dynamics, it may fall short in modeling the complex and high-dimensional variations commonly present in large-scale hydrological or climate datasets. Second, the modeling capacity of the FDTDAR framework is constrained by the number of thresholds and lag terms that can be practically specified, which limits its structural flexibility compared with more adaptive, data-driven models such as LSTM. Third, the prediction performance

of FDTDAR models often depends heavily on the correct setting of thresholds and lag structures. This adjustment process can be challenging in practice and often requires expert domain knowledge or a large number of traversal trials, which may reduce the generalization ability of the model across different regions or datasets.

In contrast, machine learning models, particularly deep learning approaches, can learn complex patterns, nonlinearities, and long-term dependencies directly from the data without strong prior assumptions about model structure (van Cranenburgh et al., 2022). These models often demonstrate outstanding predictive performance in many benchmark tasks. However, their "black-box" nature raises concerns about interpretability and transparency, which are critical in scientific and decision-making contexts such as hydrology (Beven, 2020; Hosseini et al., 2025). In addition, the high computational cost, demand for large training datasets, and sensitivity to hyperparameter tuning may restrict their applicability in resource-limited environments or real-time forecasting systems.

Given these trade-offs, this study focuses on DAR-type models as a theoretically grounded and operationally tractable alternative to traditional AR-GARCH models. While the proposed FDTDAR framework strikes a balance between model simplicity and nonlinear expressiveness, we acknowledge that future research should include comprehensive comparisons with state-of-the-art machine learning models. Such efforts would help further assess the strengths and limitations of the proposed approach and explore the potential of hybrid frameworks that combine the interpretability of statistical models with the flexibility of machine learning techniques.".

4. While the use of average interval width (AIW) and containing ratio (CR) are appropriate to assess the prediction uncertainty, the manuscript lacks detail on how prediction intervals were constructed. Clarification on whether these are based on analytical variance, bootstrapping, or Monte Carlo simulations is necessary.

**Responses:** We agree that a clear explanation of the construction method of the prediction intervals is crucial to assess the reliability of uncertainty analysis. We added explanations about AIW and CR values in the section "3.6 Comparative evaluation

methods" in the revised manuscript, but due to space limitations, we did not reflect their specific calculation formulas in the main text, but cited the reference source. The description of lines 324-326 on page 15 of the revised manuscript is: "The interval forecasting performance was evaluated using the Average Interval Width (AIW) and Coverage Rate (CR), with the calculation formulas detailed in Wang et al. (2023b).".

Wang, H., Song, S., Zhang, G., Ayantoboc, O.O., 2023b. Predicting daily streamflow with a novel multi-regime switching ARIMA-MS-GARCH model. Journal of Hydrology: Regional Studies, 47: 101374. https://doi.org/10.1016/j.ejrh.2023.101374

The AIW used in this study is expressed as RIW in this reference, which provides a detailed process for calculating RIW and CR values.

5. The analysis clearly shows that the Student's t-distribution improves predictive performance over the Gaussian assumption. The authors are encouraged to provide more discussion on how degrees of freedom were selected, and whether any skewed or generalized t-distributions were considered or could be more appropriate for heavy-tailed hydrological data.

**Responses:** Thank you for this thoughtful comment. In the revised manuscript, we have added an explanation of how the degrees of freedom parameter was selected. Specifically, the freedom was estimated jointly with other model parameters through maximum likelihood estimation. To ensure numerical stability and meaningful interpretation, we imposed a constraint that freedom must be greater than 2, which guarantees the existence of finite variance. This is described in the revised manuscript on page 13, line 288-289: "The degrees of freedom of the student's t distribution are estimated jointly with the other model parameters, and a constraint of greater than 2 is imposed to ensure numerical stability.".

Regarding considerations for skewed or generalized t-distributions, we agree that such extensions may be more appropriate for modeling asymmetric and fat-tailed behavior in hydrological data. While our current focus is on symmetric distributions, we have added a discussion in Section 5.4 (pages 27-28, lines 545-550) acknowledging this limitation and suggesting that future work could explore skewed or generalized tdistributions to further improve the flexibility and robustness of the model in capturing extreme events. The specific description is: "In terms of the model residuals, although the student's t distribution used in this study effectively captures heavy tails, it does not account for potential asymmetry in the distribution of hydrological residuals. In real-world streamflow processes, especially during flood or drought events, residuals may exhibit skewness in addition to heavy tails. Therefore, more flexible distributions, such as the skewed or generalized t-distributions, may offer a better fit for such cases. While these distributions were not explored in the present study, they represent a promising direction for future research aimed at enhancing the model's adaptability to extreme or asymmetric behavior.".

Apart from these general comments, the authors should also take into consideration a few minor points.

**Minor remarks:**

1. Terms such as FDTDAR-n and FDTDAR-t should be introduced earlier and used consistently.

**Responses:** Thank you for pointing this out. In the revised manuscript, we mentioned the expression of FDTDAR-n and FDTDAR-t models in the last paragraph of the Introduction, which are the two most important models in this paper. At the same time, we carefully checked other terms in the text to ensure that they are displayed when they are first mentioned. We also carefully reviewed the manuscript to ensure that these terms are used consistently in all subsequent sections.

2. Several grammatical and syntactic issues are present throughout the manuscript. A round of professional language editing is recommended.

**Responses:** Thanks for your suggestion. We have already asked Professor Thian Yew Gan to help us with professional revisions. Prof Gan is internationally renowned for his many innovative, multidisciplinary contributions to our understanding in hydrology, hydroclimatology, cryosphere, remote sensing of environment, and water resources management. He is a pioneer in research regarding climate change impact to water resources, and developed many practical engineering tools/models for hydrologic

forecasting, and innovative algorithms to retrieve large-scale spatial information from remotely sensed data.

3. Recent advances in time series forecasting using deep learning could be briefly referenced to contextualize the DTDAR approach.

**Responses:** Thank you for your valuable suggestion. We agree that briefly mentioning the strengths of deep learning in time series forecasting helps to better clarify the motivation and positioning of the DTDAR approach. In the revised manuscript, we have added a paragraph in the discussion section outlining the rationale for choosing DAR-type models, along with a reflection on their limitations compared to machine learning approaches. We also clarified that, although machine learning methods are powerful, they typically require large datasets, substantial computational resources, and may lack interpretability, which are especially critical in scientific and decision-making contexts such as hydrology. In addition, we objectively acknowledge that future research should include comprehensive comparisons with state-of-the-art machine learning models.